# Circulating microRNAs as biomarkers for diabetic retinopathy stage identification: A DTA systematic review and meta-analysis

Miriam Martínez-Santos[1,2,3], María Ybarra[1,2,3], Maria E. Pires[1,2,3], Chiara Ceresoni[1,2,3], Elías Martínez-López[4], Javier Sancho-Pelluz[2,3], Maria Oltra[2,3]*, Jorge M. Barcia[1,2,3]

**1** Escuela de Doctorado Universidad Católica de Valencia San Vicente Mártir, Valencia, Spain, **2** Facultad de Medicina y Ciencias de la Salud, Universidad Católica de Valencia San Vicente Mártir, Valencia, Spain, **3** Centro de Investigación Traslacional San Alberto Magno, Universidad Católica de Valencia San Vicente Mártir, Valencia, Spain, **4** Department of General and Digestive Surgery, Hospital Universitario Doctor Peset, Valencia, Spain

* maria.oltra@ucv.es

## Abstract

### Purpose

To evaluate the diagnostic accuracy of circulating miRNAs in distinguishing between different diabetic retinopathy (DR) stages in type 2 diabetes mellitus (T2DM).

### Methods

We conducted a systematic review and meta-analysis in accordance with PRISMA-DTA and Cochrane guidelines. The protocol was not registeres and no external funding was received. A comprehensive search was performed in PubMed, CENTRAL, Scopus, Web of Science, ScienceDirect, and ClinicalTrials (up to January 2025) to identify diagnostic test accuracy studies on circulating miRNAs for DR. Eligible studies included three predefined comparisons: healthy controls versus DR (CTL vs DR), T2DM without DR versus DR (T2DM vs DR), and non-proliferative versus proliferative DR (NPDR vs PDR). DR diagnosis was confirmed using fundus fluorescein angiography and/or fundus examination. Two reviewers independently conducted study selection, data extraction, and risk of bias assessment with QUADAS-2; certainty of evidence was assessed using GRADE. Data were synthesized using a bivariate random-effects meta-analysis, with subgroup analyses, meta-regression, and sensitivity analyses to explore heterogeneity. Data were synthesized via a bivariate random-effects meta-analysis, with subgroup analyses, meta-regression, and sensitivity tests to explore heterogeneity.

**Data availability statement:** All relevant data are within the manuscript and its Supporting Information files.

**Funding:** The present work received internal funds from Centro de Investigación Traslacional SanAlberto Magno (CITSAM, UCV) and external funds from Agencia Estatal de Investigación Española (PID2020-117875GB-10), Instituto de Salud Carlos III (ISCIII, PI21/00083) and the European Union research fund, HORIZON MSCA 2021-DN-01-01_RETORNA 101073316 and Generalitat Valenciana ACIF 2023-128-001. The funders had no role in the study design, data collection and analysis, decision to publish, or preparation of the manuscript.

**Competing interests:** The authors have declared that no competing interests exist.

## Results

Sixteen studies (1849 participants; 21 miRNAs) were included. For CTL vs DR (7 studies), pooled sensitivity was 77% (70–82) and specificity 84% (77–89), AUC 0.86 (0.82–0.89). For T2DM vs DR (9 studies), sensitivity was 81% (75–86) and specificity 80% (71–87), AUC 0.88 (0.84–0.91). For NPDR vs PDR (12 studies), sensitivity was 84% (79–87) and specificity 82% (76–88), AUC 0.90 (0.87–0.93). Heterogeneity arose chiefly from sample matrix, normalization strategies and inter-study expression trends. Patient selection posed the greatest bias risk.

## Conclusions

Circulating miRNAs exhibit promising diagnostic accuracy for differentiating among various stages of DR. However, future large, prospective studies in diverse populations and standardized pre-analytical protocols are required to confirm and translate these findings.

## Introduction

Diabetic retinopathy (DR) is one of the leading causes of vision loss among working-age adults worldwide [1]. As the global prevalence of type 2 diabetes mellitus (T2DM) continues to rise, so does the incidence of DR, posing a growing public health concern [2]. The condition progresses through well-defined stages, starting with non-proliferative diabetic retinopathy (NPDR) and potentially advancing to proliferative diabetic retinopathy (PDR), characterized by neovascularization and an increased risk of retinal detachment or hemorrhage [3]. The cumulative occurrence of progression from NPDR to vision-threatening complications has been estimated at approximately 14–16%. with the risk of progression to vision loss rising significantly, reaching nearly 58% [4,5]. Accurate detection and staging of DR are therefore essential for timely clinical decision-making and for preventing irreversible vision loss [4].

Current diagnostic methods for DR include fundus examination and fluorescein angiography (FA) among others, FA is the gold standard for staging due to its high sensitivity in detecting microvascular damage [6,7]. Fundus examination is more accessible but less sensitive, particularly in early disease [8]. FA, while accurate, is invasive and resource-intensive, limiting its routine use [9]. These limitations highlight the need for non-invasive, accessible biomarkers that could support early detection and improve screening and risk stratification in broader clinical settings [10,11].

Circulating microRNAs (miRNAs) have emerged as promising candidates for this role. MiRNAs are small, non-coding RNAs involved in the post-transcriptional regulation of gene expression and are detectable in various biological fluids, including serum, plasma, aqueous humor, and extracellular vesicles [12,13]. Their high stability in circulation, disease-specific expression patterns, and accessibility through non-invasive sampling make them attractive tools for biomarker discovery [11]. Although numerous studies have investigated the diagnostic potential of circulating miRNAs in

DR, marked heterogeneity in sample types, analytical methods (e.g., RT-qPCR, microarrays, NGS), and target miRNAs has limited comparability across studies [14–18].

Notably, one prior meta-analysis has reviewed the use of circulating miRNAs for DR detection [19], it did not perform any stratification by disease stage or type of control group. This represents a notable gap in the current literature, as the ability to distinguish early from advanced stages of DR is critical for clinical decision-making.

Therefore, the aim of this systematic review and meta-analysis is to evaluate the diagnostic accuracy of circulating miRNAs in DR. Specifically, we assess their performance in distinguishing between healthy controls vs DR patients, T2DM vs DR, and NPDR vs PDR stages. This stratified approach addresses a critical unmet need by systematically analyzing the diagnostic value of miRNAs across clinically relevant disease stages and control populations.

## Methods

### Search strategy

This systematic review and meta-analysis were conducted following the Preferred Reporting Items for PRISMA-Diagnostic Test Accuracy (PRISMA-DTA) [20] for further information consult (S1 and S2 Tables in S1 File). This systematic review was not registered. We performed extensive search in the following databases: PubMed, CENTRAL, Scopus, Web of Science, Science Direct, and Clinical Trials. The last update of this review was on January 20, 2025. The search strategy was designed to identify relevant studies evaluating the diagnostic accuracy of miRNAs in DR. A combination of Medical Subject Headings (MeSH) terms and free-text keywords was applied, using Boolean operators (AND, OR, NOT) to refine the search. The main topics included: DR, miRNAs, expression profiling, biomarkers, and biological sample types (serum, plasma, aqueous humor, extracellular vesicles). For the complete search strategies please refer to (S1 Text in S1 File).

### Eligibility criteria

This systematic review and meta-analysis follow the Population–Index test–Target condition (PIT) structure, as recommended by the Cochrane. We focused on (P) Population: human participants diagnosed with DR at various stages (NPDR or PDR), as well as individuals without DR, including healthy controls and patients with T2DM without DR; (I) Index test: miRNA expression levels measured in serum, plasma, aqueous humor, or extracellular vesicles, using validated techniques such as quantitative real-time (RT-qPCR), microarrays, or next-generation sequencing (NGS); and (T) Target condition: DR. We included studies published from 2014 onward, as long as they evaluated the diagnostic accuracy of miRNAs across different stages of DR confirmed using FA and fundus examination. To be eligible, studies also needed to report key accuracy metrics such as: sensitivity, specificity, or area under the curve (AUC). On the other hand, we excluded studies that focused on other types of diabetes, *in vitro* or in animal models.

### Study selection and data extraction

Two independent reviewers (M.M-S and E.M-L.) assessed studies for eligibility based on predefined inclusion and exclusion criteria. Full-text articles of potentially relevant studies were reviewed, and any discrepancies were resolved by consensus; when necessary, a third reviewer (M.O.) provided arbitration.

From each included study, we extracted data on study characteristics (author, year, country, design), population details, biological sample type, index test platform, normalization strategy, and diagnostic accuracy metrics such as: sensitivity, specificity and AUC. The complete extraction dataset is provided in S3 Table in S1 File. Information on miRNA expression trends and cut-off values was recorded when available. Duplicate records were identified and removed in EndNote X9, following the methodology described by Kwon et al. [21]. All references were managed using EndNote X9 software.

## Quality assessment

The risk of bias and applicability concerns of the included studies were assessed using the QUADAS-2 tool [22], applied independently by two reviewers across the four standard domains: patient selection, index test, reference standard, and flow/timing. Discrepancies were resolved by consensus. In addition, the GRADE approach [23], adapted for diagnostic test accuracy studies, was used to evaluate the overall strength of evidence and guide recommendations.

## Statistical analysis

We calculated 2 × 2 contingency tables true positives (TP), false positives (FP), false negatives (FN), and true negatives (TN) for each included study. Pooled sensitivity, specificity, likelihood ratios (PLR, NLR), diagnostic odds ratio (DOR), and SROC curve were estimated using a bivariate random-effects model, recommended for diagnostic test accuracy meta-analyses. This model jointly accounts for sensitivity and specificity, including their correlation and between-study variability. Since most studies did not report diagnostic thresholds, a hierarchical HSROC model was not applicable. Instead, we generated empirical SROC curves from bivariate estimates to visualize overall diagnostic performance. Heterogeneity was assessed using Cochran's Q-test and the I2 statistic derived from the bivariate model. An I2 value above 50% or a p-value < 0.05 was considered indicative of substantial heterogeneity. To investigate its potential sources, pre-specified univariable meta-regressions were performed within the bivariate framework, using variables such as sample size, country of origin, biological specimen type normalization strategy, and miRNA expression pattern. To assess the robustness of our findings, sensitivity analyses were carried out using leave-one-out methods and influence diagnostics. Model assumptions were evaluated via residual deviance plots and bivariate normality tests. Finally, to evaluate potential publication bias, Deeks' funnel plot asymmetry test was performed. Fagan nomograms were generated to translate likelihood ratios into post-test probabilities, facilitating clinical interpretation. Percentages are reported as whole numbers rounded to the nearest integer; exact estimates and 95% confidence intervals are provided in the corresponding tables and figures. All statistical analyses were performed at 95% CI, using STATA 18 (STATA Corporation, College Station, TX, USA), incorporating the MIDAS package for meta-analysis of DTA. Additional details are provided in S4, S5 and S6 Tables in S1 File.

# Results

## Study characteristics and quality assessment

A total of 454 articles were identified from various databases, including: CENTRAL (n = 2), Scopus (n = 122), PubMed (n = 77), Clinical Trials (n = 1), Web of Science (n = 125), and Science Direct (n = 127). Additionally, 6 abstracts were retrieved from grey literature (ARVO). After the removal of 270 duplicates, 180 records were screened based on title and abstract, resulting in 60 full-text articles being assessed for eligibility. Among these, 38 studies were excluded for the following reasons: *in vitro* studies (n = 8), reviews (n = 4), animal studies (n = 6), and studies with microvascular complications other than DR (n = 20). Ultimately, 16 studies met the inclusion criteria and were included in the quantitative and qualitative synthesis meta-analysis (Fig 1).

These 16 studies analyzed 1.849 patients and investigated 21 distinct miRNAs. Among these miRNAs, 6 were detected in plasma, 14 in serum, and 1 in exosomes. In terms of study design, 14 studies were case-control [14–18,24–31], while 2 were cross-sectional studies [32,33]. Geographically, the majority of studies were from China (n = 11) [14,17,18,24,25,27,28,30,33,34], followed by Egypt (n = 3) [16,26,31], Italy (n = 1) [29], and Indonesia (n = 1) [32]. Regarding the analytical approach, 3 studies assessed miRNA panels [15,27,29], whereas the remaining 13 studies focused on single miRNA analysis [14,16–18, 24–26,28,30,32–34]. In terms of comparison groups, 9 studies have a single type of comparison, whereas 7 studies had multiple comparisons across different disease stages. The distribution of comparisons was as follows: 6 studies analyzed CTL vs DR [24,25,27,32–34], 7 studies analyzed T2DM vs DR [14,16,24,26,28–30],

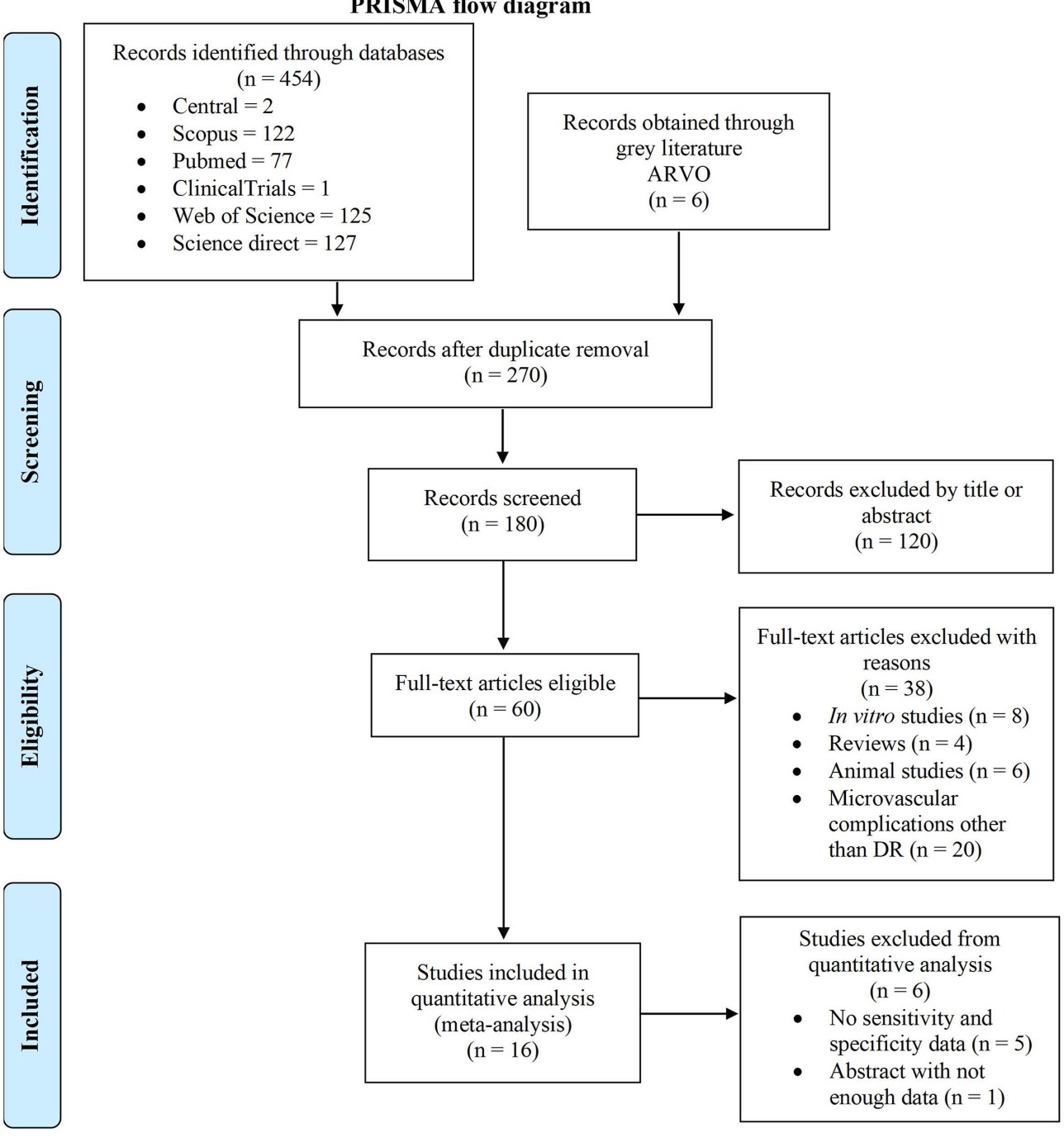

**Fig 1. Flow diagram detailing the selection of studies included in the diagnostic accuracy meta-analysis.**

**Table 1. Quantitative and qualitative characteristics of included studies.**

| Study (year) | Country | Study Design | Comparation | miRNAs | Expression | Specimen | Method | Normalization | TP | TN | FP | FN | Sen % | Spe % | Total (n) |
|---|---|---|---|---|---|---|---|---|---|---|---|---|---|---|---|
| Jiang et al., 2017 | China | Case control | CTL vs DR | miR-21 | Up | Plasma | RT-qPCR | U6 | 82 | 104 | 11 | 42 | 66.1 | 90.4 | 239 |
| Qin et al., 2017 | China | Case control | CTL vs DR | miR-126 | Down | Plasma | RT-qPCR | U6 | 32 | 53 | 6 | 7 | 81.25 | 90.34 | 98 |
| Wan et al., 2017 | China | Case control | CTL vs DR | miR-7 | Down | Serum | RT-qPCR | MIR2911 | 58 | 54 | 20 | 18 | 76 | 73 | 150 |
| Wan et al., 2017 | China | Case control | CTL vs DR | miR-7 (exosome) | Down | Exosome | RT-qPCR | MIR2911 | 57 | 57 | 17 | 18 | 75 | 77 | 149 |
| Liu et al., 2018 | China | Cross Sectional | CTL vs DR | miR-211 | Up | Serum | RT-qPCR | U6 | 54 | 29 | 4 | 10 | 85 | 87 | 97 |
| Li et al., 2019 | China | Case control | CTL vs DR | miR-4448, miR-338-3p, miR-190a-5p, mir485-5p, miR-9-5p | Up/Down | Serum | RNA-Seq | DESeq2 | 9 | 10 | 1 | 1 | 90 | 90.9 | 21 |
| Surasmiati et al., 2023 | Indonesia | Cross sectional | CTL vs DR | miR-126 | Down | Serum | RT-qPCR | miRNA-328-3p | 9 | 8 | 2 | 2 | 75 | 50 | 21 |
| Qin et al., 2017 | China | Case control | T2DM vs DR | miR-126 | Down | Plasma | RT-qPCR | U6 | 69 | 42 | 2 | 12 | 84.8 | 94.9 | 125 |
| Shaker et al., 2019 | Egypt | Case control | T2DM vs DR | miR-20b | Down | Serum | RT-qPCR | SNORD68 | 31 | 18 | 12 | 19 | 62 | 60 | 80 |
| Shaker et al., 2019 | Egypt | Case control | T2DM vs DR | miR-17-3p | Down | Serum | RT-qPCR | SNORD68 | 46 | 17 | 13 | 4 | 92 | 56.7 | 80 |
| Yin et al., 2020 | China | Case control | T2DM vs DR | miR-210 | Up | Serum | RT-qPCR | U6 | 92 | 32 | 8 | 18 | 83.6 | 80 | 150 |
| Santavito et al., 2021 | Italy | Case control | T2DM vs DR | miR-25-3p, miR-320b, miR-495-3p | Up/Down | Plasma | RT-qPCR | miR-19-5p, miR-125a-5p | 17 | 9 | 1 | 3 | 85 | 85 | 30 |
| Wang et al., 2021 | China | Case control | T2DM vs DR | miR-374a | Up | Serum | RT-qPCR | U6 | 110 | 58 | 12 | 27 | 80.3 | 82.9 | 207 |
| Saleh et al., 2022 | Egypt | Case control | T2DM vs DR | miR-93 | Down | Serum | RT-qPCR | miRNA-16 | 68 | 69 | 11 | 12 | 85 | 86 | 160 |
| Saleh et al., 2022 | Egypt | Case control | T2DM vs DR | miR-152 | Up | Serum | RT-qPCR | miRNA-16 | 68 | 58 | 22 | 12 | 85 | 72 | 160 |
| Zhao et al., 2023 | China | Case control | T2DM vs DR | miR-221-3p | Up | Serum | RT-qPCR | U6 | 110 | 84 | 18 | 48 | 69.7 | 82.3 | 260 |
| Qing et al., 2014 | China | Case control | NPDR vs PDR | miR-21, miR-181c, miR-1179 | Up | Serum | RT-qPCR | U6 | 74 | 86 | 4 | 16 | 82 | 95 | 180 |
| Jiang et al., 2017 | China | Cross Sectional | NPDR vs PDR | miR-21 | Up | Plasma | RT-qPCR | U6 | 37 | 58 | 15 | 14 | 72.5 | 79.5 | 124 |
| Shaker et al., 2019 | Egypt | Case control | NPDR vs PDR | miR-20b | Down | Serum | RT-qPCR | SNORD68 | 14 | 23 | 7 | 6 | 70 | 76.6 | 50 |
| Shaker et al., 2019 | Egypt | Case control | NPDR vs PDR | miR-17-3p | Down | Serum | RT-qPCR | SNORD68 | 10 | 24 | 6 | 10 | 50 | 80 | 50 |

*(Continued)*

**Table 1.** (Continued)

| Study (year) | Country | Study Design | Comparation | miRNAs | Expression | Specimen | Method | Normalization | TP | TN | FP | FN | Sen % | Spe % | Total (n) |
|---|---|---|---|---|---|---|---|---|---|---|---|---|---|---|---|
| Hui et al, 2019 | China | Case control | NPDR vs PDR | miR-126 | Down | Plasma | RT-qPCR | cel-miR-39-3p | 28 | 31 | 12 | 8 | 78.4 | 73.2 | 79 |
| Ma et al., 2019 | China | Case control | NPDR vs PDR | miR-93 and miR-21 | Up | Plasma | RT-qPCR | U6 | 39 | 30 | 4 | 3 | 92 | 89 | 76 |
| Ma et al., 2019 | China | Case control | NPDR vs PDR | miR-93 | Up | Plasma | RT-qPCR | U6 | 37 | 28 | 6 | 5 | 89 | 81 | 76 |
| Ma et al., 2019 | China | Case control | NPDR vs PDR | miR-21 | Up | Plasma | RT-qPCR | U6 | 38 | 24 | 10 | 4 | 90 | 71 | 76 |
| Yin et al., 2020 | China | Case Control | NPDR vs PDR | miR-210 | Up | Serum | RT-qPCR | U6 | 41 | 45 | 15 | 9 | 84.2 | 78.9 | 110 |
| Wang et al., 2021 | China | Case Control | NPDR vs PDR | miR-374a | Up | Serum | RT-qPCR | U6 | 61 | 51 | 13 | 10 | 84.2 | 78.8 | 135 |
| Saleh et al., 2022 | Egypt | Case control | NPDR vs PDR | miR-93 | Down | Serum | RT-qPCR | miRNA-16 | 34 | 25 | 15 | 6 | 85 | 63 | 80 |
| Saleh et al., 2022 | Egypt | Case control | NPDR vs PDR | miR-152 | Up | Serum | RT-qPCR | miRNA-16 | 34 | 32 | 8 | 6 | 85 | 80 | 80 |
| Salem et al., 2022 | Egypt | Case control | NPDR vs PDR | miR-181c | Up | Serum | RT-qPCR | U6 | 54 | 60 | 0 | 6 | 90 | 100 | 120 |
| Salem et al., 2022 | Egypt | Case control | NPDR vs PDR | miR-1179 | Up | Serum | RT-qPCR | U6 | 54 | 48 | 12 | 6 | 90 | 80 | 120 |

TP: True Positives, TN: True Negatives; FP: False Positives; FN: False Negatives; Sen: Sensitivity.

9 studies analyzed NPDR vs PDR [15–18,26,28,30,31,34]. The primary technique employed for miRNA detection was RT-qPCR in 15 studies, while RNA-seq was used in one study [27]. Regarding normalization methods, 9 studies utilized U6 [14,15,18,24,28,30,31,33,34], while the remaining 7 studies applied different miRNA normalization methods [16,17,25–27,29,32] (Table 1).

The QUADAS-2 assessment showed that the main sources of bias came from how patients were selected and how the index test was applied. A high risk of bias was mostly linked to patient selection, often due to non-random sampling methods and retrospective study designs. In the index test domain, there were some concerns as well, particularly in studies that didn't clearly define diagnostic thresholds ahead of time. On the other hand, the reference standard and the flow and timing of the studies generally showed a lower risk of bias, with most studies following proper diagnostic procedures and reasonable timelines. When it came to applicability, most studies posed low concern across all domains. However, a few showed minor issues related to the index test, mainly because of differences in how it was applied across studies. A detailed summary of the risk of bias and applicability concerns is shown in (Fig 2).

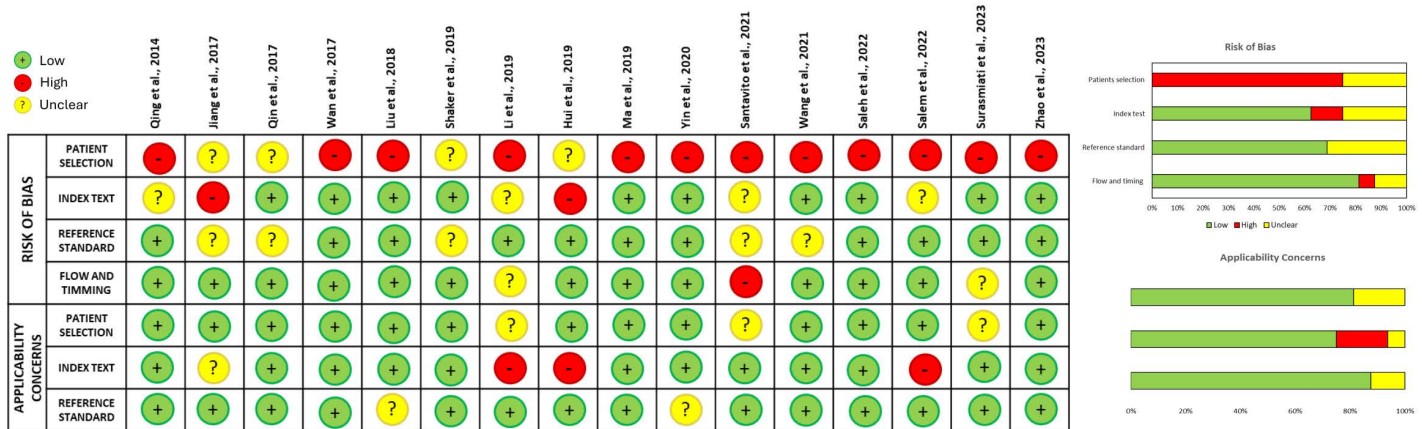

**Fig 2. Risk of bias and applicability assessment using the QUADAS-2 tool.**

## Diagnostic accuracy of miRNAs in CTL vs DR, T2DM vs DR, and NPDR vs PDR comparisons

A total of 7 studies contributed data to the comparison of CTL vs DR. The pooled estimates from the random-effects model showed a summary sensitivity of 77% (70–82), with an I2 of 47%, and a summary specificity of 84% (77–89), with an I2 of 62%, indicating moderate heterogeneity (Fig 3A-B). The SROC curve yielded an AUC of 0.86 (0.84–0.92) (Fig 3C), suggesting moderate-to-high overall accuracy. To understand the clinical relevance of these findings, a Fagan nomogram was constructed using the actual pre-test probability of DR in the population studied 22% [2]. The plot revealed a positive miRNA result raises the probability of having the disease to 58%, while a negative result reduces it to just 7% (Fig 4A). In parallel, a scatter matrix was used to visualize the relationship between the likelihood ratios across all miRNA tests included in this comparison. The pooled positive likelihood ratio (PLR) was 4.77 (3.19–7.13), indicating that patients with DR are nearly five times more likely to test positive than those without the disease. On the other hand, the pooled negative likelihood ratio (NLR) was 0.29 (0.23–0.37), suggesting a notable reduction in the probability of disease following a negative result (Fig 4A-B). Lastly, Deeks' test indicated no significant presence of publication bias (p = 0.27) (Fig 5A).

Data from 9 studies in T2DM vs DR comparison indicated a summary sensitivity of 81% (75–86), with a specificity of 80% (71–87), with an I2 of 73%, suggesting moderate to high heterogeneity in both measures (Fig 3D-E). The SROC curve reported an AUC of 0.88 (0.80–0.95), indicating strong overall diagnostic accuracy (Fig 3F). Fagan nomogram was constructed using the actual pre-test probability of DR in this population 28% [35]. The plot showed that a positive miRNA result would increase the probability of detecting DR of 61%, while a negative result would lower the likelihood to just 8%. This reflects a meaningful shift in post-test probabilities, reinforcing the role of miRNAs in helping clinicians differentiate between uncomplicated diabetes and the onset of DR (Fig 4C). The scatter matrix illustrated the distribution of diagnostic performance showed the PLR was 4.1 (2.8–6.1), suggesting that patients with DR are approximately four times more likely to test positive compared to those with T2DM alone. Meanwhile, the NLR was 0.23 (0.17–0.32), indicating a substantial decrease in the probability of disease following a negative test (Fig 4D). The Deeks' funnel plot analysis showed no significant evidence of publication bias (p = 0.81), indicating a low risk that the results are influenced by publication bias (Fig 5B).

Twelve studies were included in this analysis comparing NPDR vs PDR. The meta-analysis found a pooled sensitivity of 84% (79–87) with an I2 of 56% and a specificity of 82% (76–88) with an I2 of 77%, showing moderate-to-high heterogeneity, (Fig 3G-H). The SROC curve demonstrated an AUC of 0.90 (0.82–0. 91), pointing to high overall diagnostic accuracy (Fig 3I). The Fagan nomogram was constructed using a pre-test probability of 17% [5], based on the actual prevalence observed in this population.

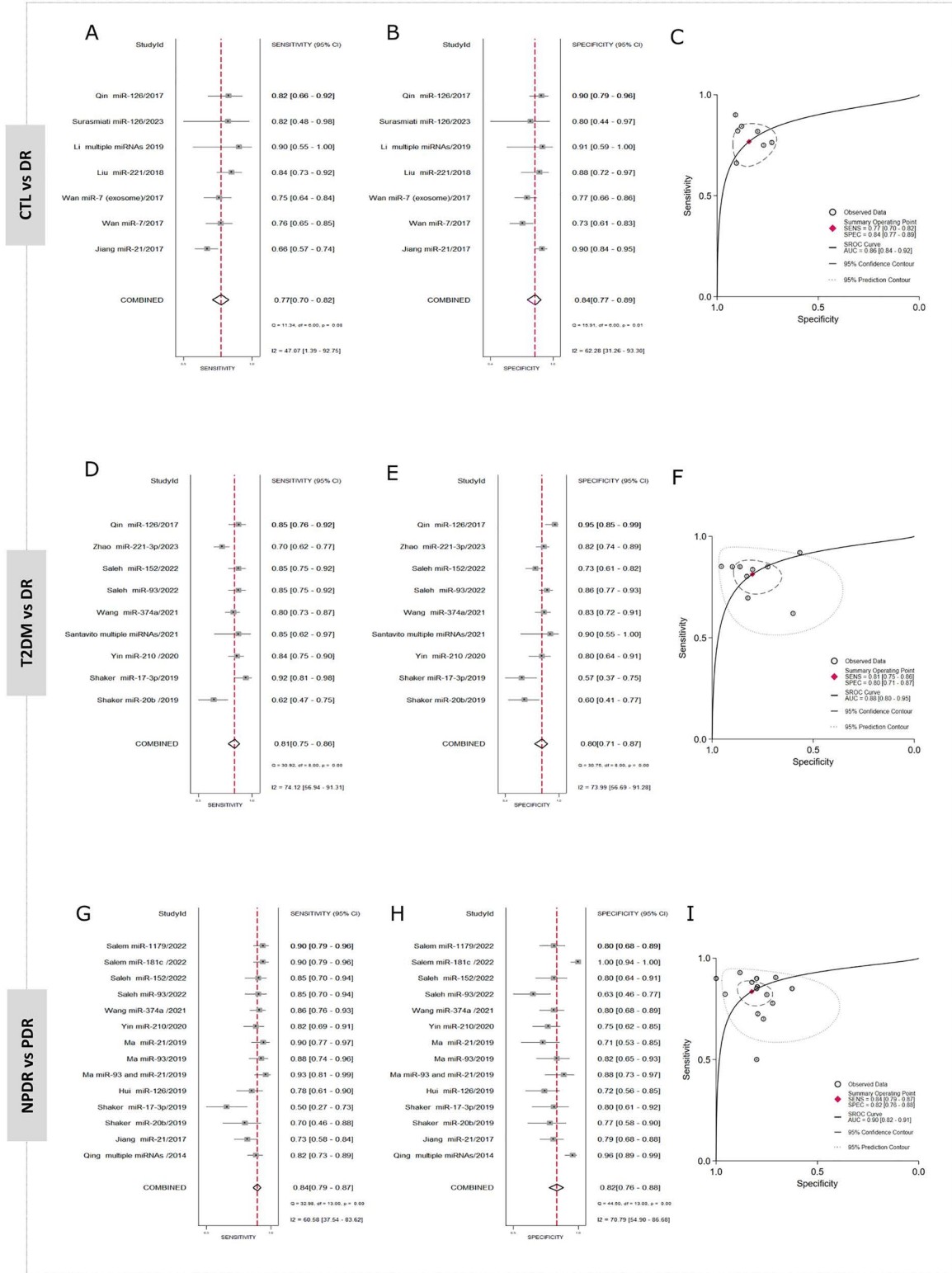

**Fig 3. Forest plots of sensitivity and specificity and SROC curves showing diagnostic accuracy of miRNAs: CTL vs DR (A-C), T2DM vs DR (D-F), and NPDR vs PDR (G-I) comparisons.**

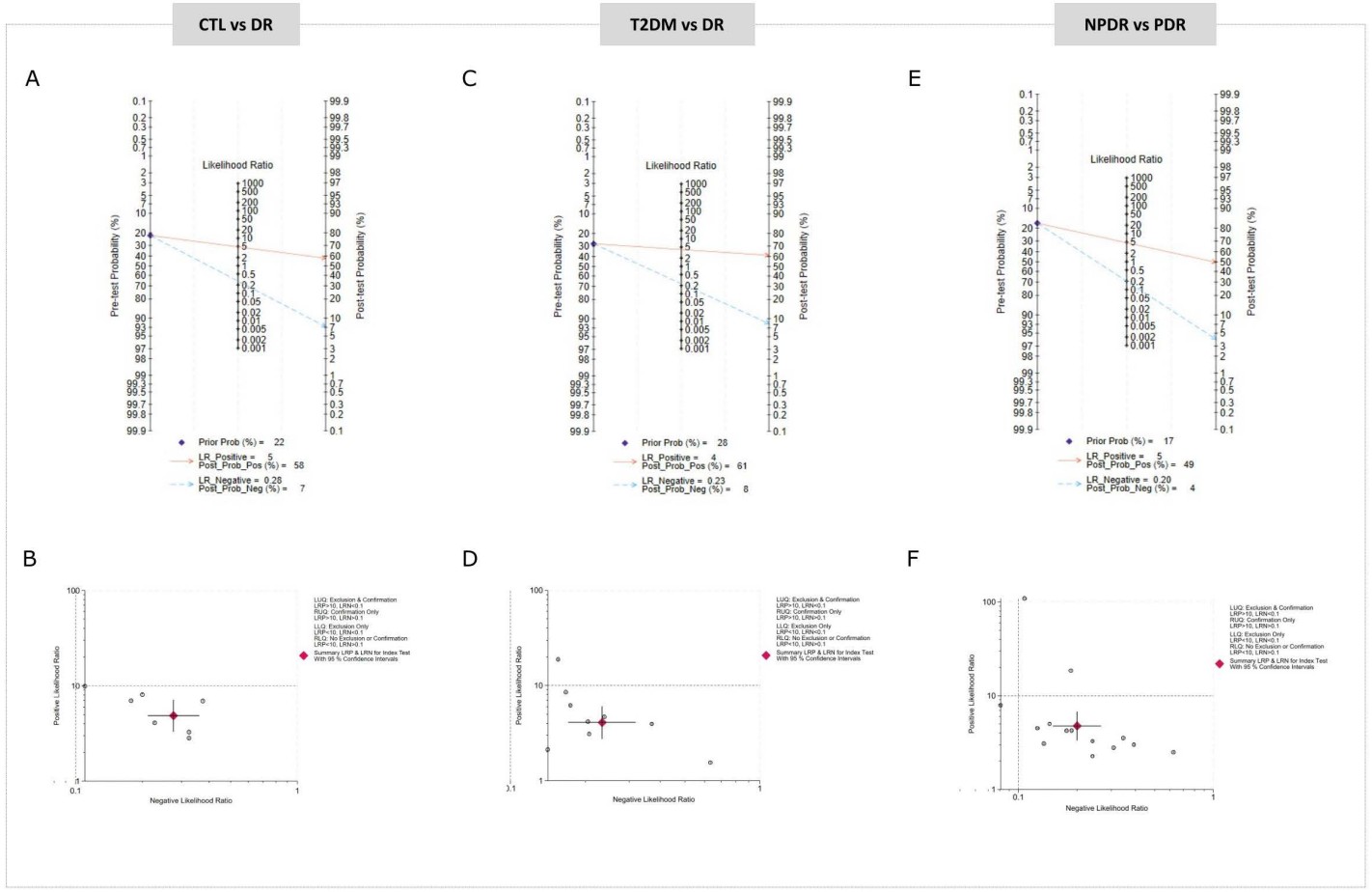

**Fig 4. Clinical applicability of miRNAs for the diagnosis of DR.** Fagan's nomogram illustrates post-test probabilities of miRNA-based diagnosis and Summary plot PLR and NLR: CTL vs DR(A-B), T2DM vs DR(C-D) and NPDR vs PDR (E-F).

The analysis revealed that a positive miRNA result increases the likelihood of diagnosing PDR to 49% (Fig 4E). A scatter matrix showed that PLR was 4.8 (3.3–6.8), indicating that patients with PDR are nearly five times more likely to test positive compared to those with NPDR. Meanwhile, the NLR was 0.20 (0.15–0.27), showing that a negative result significantly reduces the chance of PDR (Fig 4F). Finally, the Deeks' funnel plot showed no clear evidence of publication bias (p=0.08) (Fig 5C).

### Subgroup analysis and meta-regression in CTL vs DR, T2DM vs DR, and NPDR vs PDR comparisons

Studies conducted in China demonstrated higher diagnostic performance across all three comparison groups, as reflected by AUC. Specifically, the AUC reached 0.86 for the CTL vs DR group, 0.87 for T2DM vs DR, and 0.90 for NPDR vs PDR, indicating stronger diagnostic accuracy in this population (Tables 2-4).

When it came to the type of biological sample used for miRNA detection, some differences emerged. In the CTL vs DR group (Table 2), serum samples performed better than plasma. Serum-based tests showed a sensitivity of 80% and a specificity of 82%. These results suggest that serum may offer an edge for detecting DR in this group. Interestingly in the T2DM vs DR group, where plasma outperformed serum, achieving a higher AUC of 0.93 compared to 0.85, showing better diagnostic precision (Table 3). Similarly, in the NPDR vs PDR group, plasma again came out on top, with an AUC of 0.89 and a DOR of 25.83 (Table 4), reinforcing its utility in detecting more advanced disease stages.

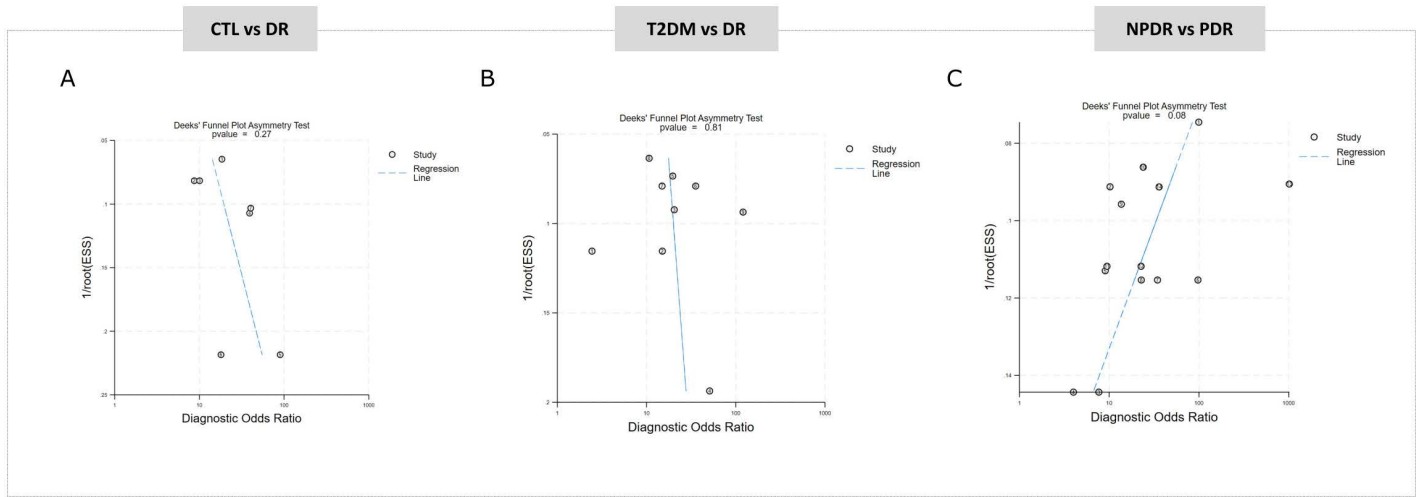

**Fig 5. Deeks' funnel plots assessing publication bias for miRNA diagnostic studies across three comparison groups: CTL vs DR (A), T2DM vs DR (B), and NPDR vs PDR (C).**

**Table 2. Summary of diagnostic performance of miRNAs based on subgroup analysis in CTL vs DR.**

| | Subgroup analysis (CTL vs DR) | | | | | |
|---|---|---|---|---|---|---|
| | Sen (95% CI) | Spe (95% CI) | PLR (95% CI) | NLR (95% CI) | DOR (95% CI) | AUC (95% CI) |
| **Origin of the study** | | | | | | |
| China (n=6) | 0.75 (0.70-0.79) | 0.84 (0.79-0.87) | 4.89 (3.14-7.61) | 0.28 (0.22-0.38) | 17.77 (9.99-31.63) | 0.86 |
| Rest of the world (n=1) | 0.86 | 0.75 | / | / | / | 0.56 |
| **Specimen** | | | | | | |
| Plasma (n=2) | 0.69 (0.62-0.76) | 0.87 (0.80-0.92) | 5.44 (3.46-8.56) | 0.31 (0.17-0.55) | 16.74 (8.97-31.24) | |
| Serum (n=4) | 0.80 (0.73-0.86) | 0.82 (0.76-0.88) | 5.60 (2.31-13.61) | 0.23 (0.15-0.35) | 27.20 (7.31-101.23) | 0.85 |
| Exosome (n=1) | 0.75 | 0.77 | / | / | / | 0.77 |
| **Sample size** | | | | | | |
| >100 (n=4) | 0.74 (0.69-0.78) | 0.82 (0.78-0.86) | 4.50 (2.71-7.48) | 0.30 (0.22-0.41) | 15.84 (7.79-32.20) | 0.84 |
| <100 (n=3) | 0.83 (0.71-0.91) | 0.84 (0.73-0.92) | 5.33 (2.89-9.82) | 0.20 (0.11-0.35) | 26.89 (10.04-72.05) | 0.91 |
| **Expression** | | | | | | |
| Up (n=2) | 0.72 (0.65-0.78) | 0.90 (0.84-0.94) | 7.52 (3.54-15.99) | 0.26 (0.11-0.61) | 30.16 (6.26-145.23) | |
| Down (n=4) | 0.78 (0.72-0.83) | 0.80 (0.74-0.85) | 4.22 (2.45-7.24) | 0.27 (0.19-0.36) | 17.35 (7.01-42.94) | 0.83 |
| Up/Down (n=1) | 0.90 | 0.91 | / | / | / | 0.90 |
| **miRNA Biomarker Profiling** | | | | | | |
| Single miRNA (n=6) | 0.74 (0.70-0.79) | 0.83 (0.79-0.87) | 4.63 (3.06-7.01) | 0.29 (0.23-0.37) | 16.40 (9.78-27.79) | 0.86 |
| Combination miRNAs (n=1) | 0.90 | 0.91 | / | / | / | / |
| **Normalization method** | | | | | | |
| U6 (n=3) | 0.74 (0.67-0.78) | 0.89 (0.84-0.93) | 7.23 (4.78-10.93) | 0.25 (0.14-0.43) | 25.45 (14.64-44.20) | 0.92 |
| Others (n=4) | 0.76 (0.70-0.82) | 0.76 (0.69-0.82) | 3.14 (2.37-4.16) | 0.31 (0.23-0.41) | 10.35 (6.25-17.12) | 0.83 |

**Table 3. Summary of diagnostic performance of miRNAs based on subgroup analysis in T2DM vs DR.**

| | Subgroup analysis (T2DM vs DR) | | | | | |
| --- | --- | --- | --- | --- | --- | --- |
| | Sen (95% CI) | Spe (95% CI) | PLR (95% CI) | NLR (95% CI) | DOR (95% CI) | AUC (95% CI) |
| **Origin of the study** | | | | | | |
| China (n=4) | 0.78 (0.75-0.82) | 0.84 (0.80-0.88) | 4.76 (3.18-7.12) | 0.23 (0.16-0.35) | 21.45 (10.12-45.48) | 0.87 |
| Rest of the world (n=5) | 0.82 (0.77-0.86) | 0.74 (0.68-0.79) | 2.97 (1.77-4.98) | 0.23 (0.12-0.45) | 13.94 (4.85-40.04) | 0.86 |
| **Specimen** | | | | | | |
| Serum (n=7) | 0.78 (0.75-0.82) | 0.77 (0.73-0.81) | 3.30 (2.23-4.68) | 0.26 (0.18-0.37) | 13.58 (7.57-24.36) | 0.85 |
| Plasma (n=2) | 0.85 (0.76-0.91) | 0.94 (0.84-0.98) | 14.26 (4.75-42.76) | 0.15 80.09-0.25) | 93.83 (25.58-344.24) | / |
| **Sample size** | | | | | | |
| >100 (n=6) | 0.80 (0.76-0.83) | 0.82 (0.78-0.86) | 4.51 (3.26-6.23) | 0.22 (0.16-0.30) | 21.14 (12.43-35.66) | 0.88 |
| <100 (n=3) | 0.78 (0.69-0.85) | 0.62 (0.50-0.74) | 2.02 (1.23-3.32) | 0.26 (0.08-0.83) | 9.72 (1.79-52.69) | 0.83 |
| **Expression** | | | | | | |
| Up (n=4) | 0.78 (0.74-0.81) | 0.79 (0.74-0.83) | 3.73 (2.97-4.70) | 0.25 (0.18-0.35) | 14.86 (10.24-21.55) | 0.86 |
| Down (n=4) | 0.82 (0.76-0.86) | 0.79 (0.72-0.85) | 3.83 (1.52-9.65) | 0.22 (0.10-0.51) | 18.82 (3.75-94.30) | 0.89 |
| Up/Down (n=1) | 0.85 | 0.85 | / | / | / | 0.93 |
| **miRNA Biomarker Profiling** | | | | | | |
| Single miRNA (n=8) | 0.79 (0.76-0.82) | 0.79 (0.75-0.83) | 3.65 (2.47-5.38) | 0.24 (0.17-0.34) | 16.34 (8.72-30.64) | 0.87 |
| Combination miRNAs (n=1) | 0.85 | 0.85 | / | / | / | 0.93 |
| **Normalization method** | | | | | | |
| U6 (n=4) | 0.78 (0.73-0.82) | 0.85(0.79-0.89) | 5.32 (2.98-9.49) | 0.22 (0.14-0.37) | 26.56 (9.49-70.38) | 0.87 |
| Others (n=5) | 0.82(0.77-0.86) | 0.74 (0.68-0.79) | 2.97 (1.77-4.98) | 0.23 (0.12-0.45) | 13.94 (4.85-40.04) | 0.86 |

**Table 4. Summary of diagnostic performance of miRNAs based on subgroup analysis in NPDR vs PDR.**

| | Subgroup analysis (NPDR vs PDR) | | | | | |
| --- | --- | --- | --- | --- | --- | --- |
| | Sen (95% CI) | Spe (95% CI) | PLR (95% CI) | NLR (95% CI) | DOR (95% CI) | AUC (95% CI) |
| **Origin of the study** | | | | | | |
| China (n=8) | 0.83 (0.79-0.87) | 0.81 (0.77-0.85) | 4.29 (3.05-6.05) | 0.20 (0.15-0.27) | 23.81 (12.93-43.84) | 0.90 |
| Rest of the world (n=6) | 0.83 (0.78-0.87) | 0.80 (0.75-0.85) | 3.49 (2.09-5.80) | 0.22 (0.12-0.42) | 17.45 (6.60-46.12) | 0.87 |
| **Specimen** | | | | | | |
| Serum (n=9) | 0.83 (0.79-0.86) | 0.83 (0.80-0.86) | 4.22 (2.72-6.55) | 0.22 (0.15-0.33) | 22.11 (10.20-43.68) | 0.88 |
| Plasma (n=4) | 0.87 (0.81-0.92) | 0.77 (0.70-0.84) | 3.83 (2.49-5.89) | 0.16 (0.09-0.29) | 25.83 (9.93-67.13) | 0.89 |
| **Sample size** | | | | | | |
| >100 (n=6) | 0.84 (0.80-0.87) | 0.85 (0.81-0.88) | 5.39 (3.09-9.40) | 0.19 (0.13-0.27) | 31.99 (13.34-76.72) | 0.90 |
| <100 (n=8) | 0.83 (0.78-0.87) | 0.76-0.70-0.81) | 3.24 (2.49-4.21) | 0.23 (0.14-0.39) | 15.04 (8.00-28.30) | 0.84 |
| **Expression** | | | | | | |
| Up (n=9) | 0.86 (0.83-0.89) | 0.84 (0.81-0.88) | 5.12 (3.45-7.60) | 0.16 (0.13-0.20) | 36.32 (20.50-64.37) | 0.92 |
| Down (n=4) | 0.74 (0.65-0.81) | 0.72 (0.63-0.79) | 2.54 (1.92-3.36) | 0.38 (0.24-0.61) | 7.46 (4.21-13.23) | 0.79 |
| **miRNA Biomarker Profiling** | | | | | | |
| Single miRNA (n=12) | 0.83 (0.80-0.86) | 0.83 (0.79-0.85) | 4.22 (3.09-5.78) | 0.21 (0.15-0.30) | 22.37 (12.24-40.91) | 0.86 |
| Combination miRNAs (n=2) | 0.85 (0.75-0.92) | 0.71 (0.60-0.80) | 2.97 (1.58-5.57) | 0.21 (0.12-0.36) | 14.22 (6.04-33.48) | / |
| **Normalization method** | | | | | | |
| U6 (n=9) | 0.85 (0.82-0.88) | 0.84 (0.81-0.87) | 4.97 (3.36-7.33) | 0.17 (0.13-0.23) | 33.22 (17.42-63.36) | 0.92 |
| Others (n=5) | 0.76 (0.69-0.83) | 0.73 (0.660.80) | 2.76 (2.14-3.56) | 0.33 (0.21-0.53) | 9.26 (5.51-15.57) | 0.81 |

Sample size also had an impact. In the T2DM vs DR and NPDR vs PDR groups, studies with more than 100 participants reported higher AUCs 0.88 and 0.90, respectively with improved sensitivity and specificity (Tables 3-4).

Changes in miRNA expression levels were also an important key. In the CTL vs DR group, upregulated miRNAs were associated with higher specificity 90% and a high DOR of 30.16. In contrast, downregulated miRNAs performed best in the T2DM vs DR group, with an AUC of 0.89 and a DOR of 18.82 (Table 3). For the NPDR vs PDR group, the pattern reversed again, upregulated miRNAs provided the strongest diagnostic performance, with a sensitivity of 86%, specificity of 84%, an AUC of 0.92, and the highest DOR observed in all comparisons: 36.32 (Table 4).

Finally, the normalization strategy used in miRNA expression analysis is also a critical factor. In the three comparison groups, U6 as the internal control had a higher AUC value 0.92 in CTL vs DR, 0.87 in T2DM vs DR, and 0.92 in NPDR vs PDR pointed its reliability and consistency in these types of studies (Tables 2-4).

Metaregressions were conducted to assess the impact of methodological variables on diagnostic sensitivity and specificity across the three comparisons. Sample size, expression and normalization strategy were significantly associated with sensitivity in all groups (Fig 6A-C) while sample size also influenced specificity in both the CTL vs DR and T2DM vs DR comparisons (Fig 6A,C). The country of origin had an effect on sensitivity in the T2DM vs DR and NPDR vs PDR groups, and also on specificity in the latter (Fig 6B,C). Specimen type was a relevant factor for sensitivity in T2DM vs DR and NPDR vs PDR, and for specificity in the NPDR vs PDR group (Fig 6B,C).

## Sensitivity analysis CTL vs DR, T2DM vs DR and NPDR vs PDR

For evaluating stability of our findings, sensitivity analysis was performed by assessing the effect of single studies and detecting potential outliers. Good fit of the model to the CTL vs DR group was demonstrated by the residual deviance plot (Fig 7A). Bivariate normality between the TP and TN distribution was confirmed (p = 0.555 and p = 0.655, respectively) (Fig 7B). Jiang et al. (miR-21) [34] was found to be of high influence by influence analysis using Cook's distance, however, its removal resulted in minimal change to the pooled estimates: sensitivity was improved from 77% to 80%, and specificity decreased from 84% to 80%. Despite this, heterogeneity (I2) was stable at 471%, indicating minimal impact on total variability (Fig 7C). No outliers were detected using standardized residuals, affirming homogeneity of the dataset (Fig 7D).

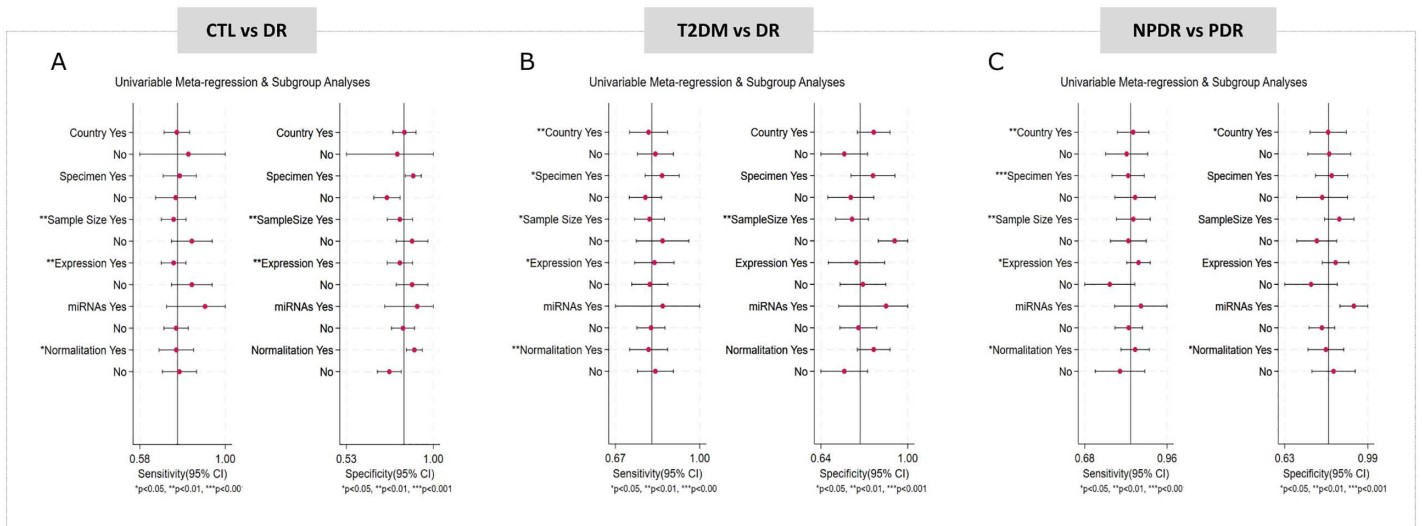

**Fig 6. Meta-regression analysis of sensitivity and specificity of miRNAs for the diagnosis of diabetic retinopathy across: CTL vs DR (A), T2DM vs DR (B), and NPDR vs PDR(C).**

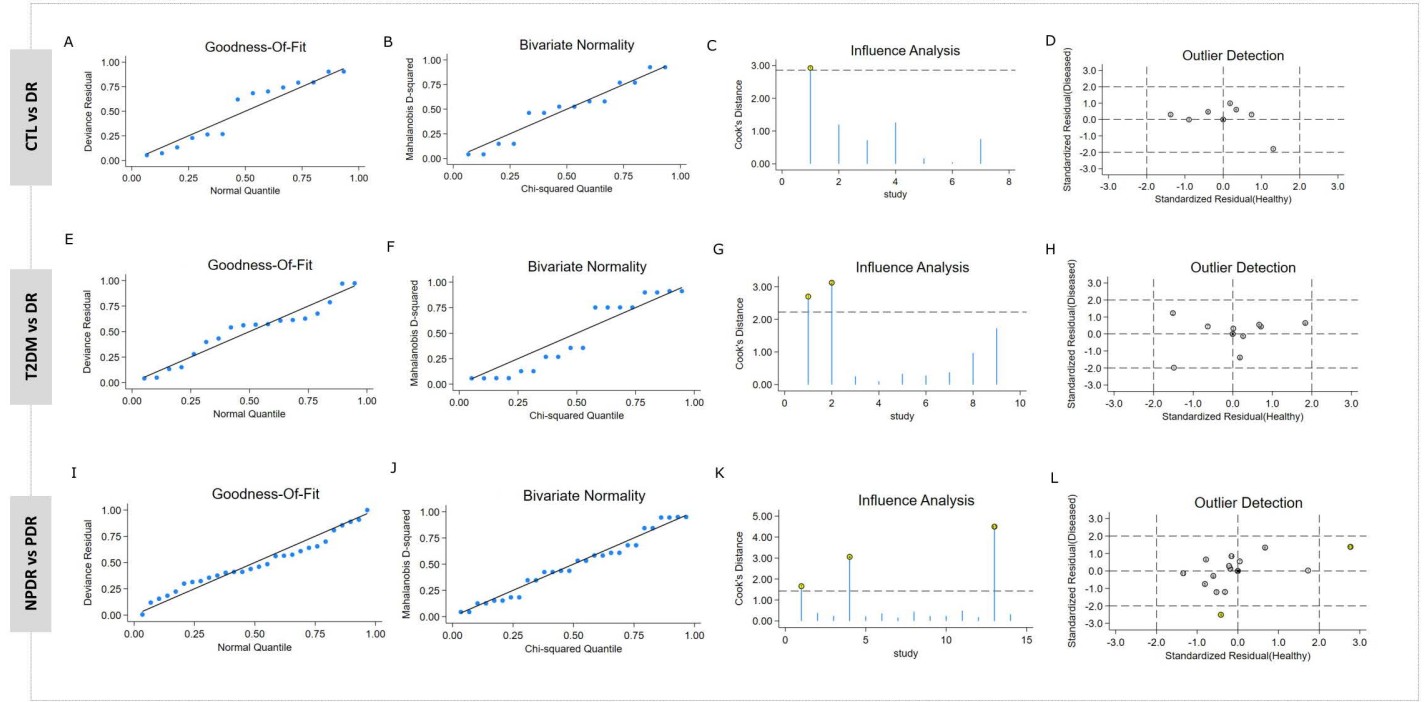

**Fig 7. Sensitivity analysis.** Goodness-of-fit (A,E,I), Bivariate normality (B,F,J) Influence analysis (C,G,K) and Outlier detection (D,H,L) across: CTL vs DR, T2DM vs DR and NPDR vs PDR.

Model adequacy was confirmed using residual deviance between T2DM and DR patients, (Fig 7E), and bivariate normality held for the distribution of diagnostic outcomes (p = 0.56 and p = 0.66) (Fig 7F). Cook's distance analysis identified two influential miRNA-specific entries from Shaker et al. (miR-20b and miR-17-3p) (Fig 7G). Removing these two miRNAs led to only marginal changes: pooled sensitivity declined from 81% to 79.%, and specificity increased from 80% to 81%. Interestingly, I2 rose notably from 74% to 97%, suggesting that these studies contributed to controlled variance across studies. Outlier assessment confirmed no excessive deviations (Fig 7H).

The residual deviance plot indicated adequate model performance in NPDR vs PDR group (Fig 7I), and bivariate normality was satisfied for the data distribution (Fig 7J). Cook's distance highlighted Qing et al. (miR-21, miR-181c, miR-1179) [15], Shaker et al. (miR-20b) [26], and Saleh et al. (miR-152) [16] as having high influence (Fig 7K). In parallel, standardized residuals identified Shaker et al [26] and Saleh et al [16] as potential outliers (Fig 7L). Excluding these studies led to slight adjustments: sensitivity increased from 84% to 85%, and specificity declined from 82% to 81.%. I2 remained constant at 71%, indicating little effect on total variability.

## Quality of the evidence (GRADE)

Across all three comparisons (CTL vs DR; T2DM vs DR and NPDR vs PDR), the quality of evidence was primarily affected by high risk of bias due to study designs of case-control and cross-sectional, as assessed by QUADAS-2. In addition, severe indirectness was always present due to variability of sample types (plasma vs. serum) and modes of miRNA expressions (overexpression vs underexpression), potentially limiting result applicability. Imprecision was not considered to be a major problem, as the confidence intervals of the pooled sensitivity and specificity were always narrow, and the sample sizes were adequate. Additionally, no publication bias evidence was found for any comparison by Contour-Enhanced Funnel Plot analysis. Despite these limitations, accuracy measures among all three groups justify conditional

recommendations on the use of miRNA profiling to distinguish CTL vs DR. T2DM vs DR, and NPDR vs PDR (Table 5-7) and low to moderate quality.

## Discussion

### Comparison with previous meta-analyses

To date, only two prior systematic reviews with meta-analytic components have investigated the diagnostic role of miRNAs in DR. Zhou et al. (2020) [36] primarily focused on microarray-based miRNA expression profiling in DR patients. Despite the term "meta-analysis" in the title, the authors did not perform a quantitative synthesis of diagnostic accuracy metrics. Instead, the work constitutes a qualitative summary of expression profiles without estimating pooled sensitivity, specificity, or AUC values. Furthermore, the study lacks a formal risk of bias assessment (e.g., QUADAS-2) and does not explore heterogeneity thereby undermining both its methodological robustness and reproducibility.

A more rigorous effort was made by Ma et al. (2022) [19], who conducted a diagnostic test accuracy meta-analysis with pooled estimates of sensitivity and specificity. The authors applied appropriate quality assessments, explored heterogeneity, and performed sensitivity analyses. However, their design presents a critical limitation: healthy individuals and T2DM were grouped together as controls. This design choice impairs the interpretability and translational potential of their findings, as it conflates physiologically distinct populations with different baseline risks and expression profiles. In contrast, our meta-analysis offers several methodological and clinical advantages. By stratifying the analysis into three distinct and clinically relevant comparison groups—healthy controls CTL vs DR, T2DM vs DR, and NPDR vs PDR, we were able to reduce inter-study heterogeneity and increase the precision and relevance of pooled diagnostic accuracy estimates comparing with Ma et al (2022). Additionally, our study uniquely integrates the GRADE framework to evaluate the strength of evidence, enhancing transparency in the interpretation of findings and the formulation of clinical recommendations.

**Table 5. Certainty of evidence for the diagnostic accuracy of miRNAs according to the GRADE approach in CTL vs DR.**

Pretest probability (global prevalence of DR): 22.27% $
Pooled sensitivity: 75% IC95% (0.71–0.79)
Pooled specificity: 84% IC95% (0.80–0.87)

**CTL vs DR**

| Outcome | Number of studies | Study design | Risk of bias | Indirect evidence | Inconsistency | Imprecision | Publication bias | Effect x1000* | Quality |
|---|---|---|---|---|---|---|---|---|---|
| True positives | 6 (626 patients) | Cross-sectional (n = 2) Case-control (n = 4) | High[1] | Serious [2] | Not serious[3] | Not serious[4] | Not detected[5] | 168 (16.76%) | ⊕⊕⊕○ Moderate |
| True negatives | 6 (626 patients) | Cross-sectional (n = 2) Case-control (n = 4) | High[1] | Serious [2] | Not serious[3] | Not serious[4] | Not detected[5] | 651 (65.12%) | ⊕⊕⊕○ Moderate |
| False Positives | 6 (626 patients) | Cross-sectional (n = 2) Case-control (n = 4) | High[1] | Serious [2] | Not serious[3] | Not serious[4] | Not detected[5] | 126 (12.61%) | ⊕⊕○○ Low |
| False negatives | 6 (626 patients) | Cross-sectional (n = 2) Case-control (n = 4) | High[1] | Serious [2] | Not serious[3] | Not serious[4] | Not detected[5] | 55 (5.51%) | ⊕⊕○○ Low |

*Number of patients per 1000 tested for a prevalence of 22.27%

1: Calculated applying QUADAS-2.

2: The evidence was rated as serious due to variability in sample types (plasma vs. serum) and differences in miRNA expression patterns (overexpression vs. underexpression), which may affect the applicability of the results

3: Heterogeneity was rated as moderate, with an I2 of 47.07% for sensitivity and 62.28% for specificity, indicating substantial variability among studies.

4: Imprecision was rated as not serious due to an adequate sample size (n = 626) and relatively narrow confidence intervals (Sensitivity: 0.71–0.79, Specificity: 0.80–0.87), indicating stable estimates.

5: No evidence of publication bias was detected based on the Contour-Enhanced Funnel Plot analysis

$: Cheung, N., Bikbov, M. M., Wang, Y. X., Tang, Y., Lu, Y., Wong, I. Y., Ting, D. S. W., Tan, G. S. W., Jonas, J. B., Sabanayagam, C., Wong, T. Y., & Cheng, C. Y. (2021). Global Prevalence of Diabetic Retinopathy and Projection of Burden through 2045: Systematic Review and Meta-analysis. *Ophthalmology*, *128*(11).

**Table 6. Certainty of evidence for the diagnostic accuracy of miRNAs according to the GRADE approach in T2DM vs DR.**

Pretest probability (global prevalence of DR in T2DM patients): 28.41%[$]
Pooled sensitivity: 80% IC95% (0.77–0.83)
Pooled specificity: 80% IC95% (0.76–0.83)

| T2DM vs DR | | | | | | | | | |
|---|---|---|---|---|---|---|---|---|---|
| Outcome | Number of studies | Study design | Risk of bias | Indirect evidence | Incon-sistency | Impre-cision | Publica-tion bias | Effect x1000* | Quality |
| True positives | 7 (1012 patients) | Case-control (n = 7) | High | Serious[2] | Serious[3] | Not serious | Not detected | 227 (22.7%) | ⊕⊕◯◯ Low |
| True negatives | 7 (1012 patients) | Case-control (n = 7) | High | Serious[2] | Serious[3] | Not serious | Not detected | 570 (57%) | ⊕⊕⊕◯ Moderate |
| False Positives | 7 (1012 patients) | Case-control (n = 7) | High | Serious[2] | Serious[3] | Not serious | Not detected | 146 (14.6%) | ⊕⊕◯◯ Very low |
| False negatives | 7 (1012 patients) | Case-control (n = 7) | High | Serious[2] | Serious[3] | Not serious | Not detected | 57 (5.7%) | ⊕⊕⊕◯ Moderate |

*Number of patients per 1000 tested for a prevalence of 28.41%

2: The evidence was rated as serious due to variability in sample types (plasma vs. serum) and differences in miRNA expression patterns (overexpression vs. underexpression), which may affect the applicability of the results.

3: Heterogeneity was rated as serious due to high inconsistency, with an I2 of 72.5% for sensitivity and 73.1% for specificity, indicating substantial variability among studies.

$: Hashemi, H., Rezvan, F., Pakzad, R., Ansaripour, A., Heydarian, S., Yekta, A., … Khabazkhoob, M. (2021). Global and Regional Prevalence of Diabetic Retinopathy; A Comprehensive Systematic Review and Meta-analysis. *Seminars in Ophthalmology*, 37(3), 291–306.

**Table 7. Certainty of evidence for the diagnostic accuracy of miRNAs according to the GRADE approach in NPDR vs PDR.**

Pretest probability (global prevalence of PDR): 17%[$]
Pooled sensitivity: 84% IC95% (0.81–0.86)
Pooled specificity: 82% IC95% (0.79–0.85)

| NPDR vs PDR | | | | | | | | | |
|---|---|---|---|---|---|---|---|---|---|
| Outcome | Number of studies | Study design | Risk of bias | Indirect evidence | Inconsis-tency | Imprecision | Publication bias | Effect x1000* | Quality |
| True positives | 9 (954 patients) | Case-control (n = 9) | High | Serious[2] | Serious[3] | Not serious | Not detected | 142 (14.2%) | ⊕⊕◯◯ Low |
| True negatives | 9(954 patients) | Case-control (n = 9) | High | Serious[2] | Serious[3] | Not serious | Not detected | 678 (67.8%) | ⊕⊕⊕◯ Moderate |
| False Positives | 9 (954 patients) | Case-control (n = 9) | High | Serious[2] | Serious[3] | Not serious | Not detected | 152 (15.2%) | ⊕◯◯◯ Very low |
| False negatives | 9 (954 patients) | Case-control (n = 9) | High | Serious[2] | Serious[3] | Not serious | Not detected | 28 (2.8%) | ⊕⊕⊕◯ Moderate |

*Number of patients per 1000 tested for a prevalence of 17%

2: The evidence was rated as serious due to variability in sample types (plasma vs. serum) and differences in miRNA expression patterns.

3: Heterogeneity was rated as serious due to high inconsistency, with an I2 of 55.6% for sensitivity and 77.1% for specificity, indicating substantial variability among studies.

$: Yang QH, Zhang Y, Zhang XM, Li XR. Prevalence of diabetic retinopathy, proliferative diabetic retinopathy and non-proliferative diabetic retinopathy in Asian T2DM patients: a systematic review and Meta-analysis. Int J Ophthalmol. 2019 Feb 18;12(2):302–311

## Biological insights and candidate miRNAs

Among the studies included, several miRNAs appeared recurrently, highlighting their potential relevance across different experimental contexts. The most frequently reported was miR-126 [17,24,32], identified in three separate studies. This miRNA has

been linked to DR through its role in vascular integrity and angiogenesis regulation, with decreased levels observed in plasma and vitreous samples of affected patients [37]. miR-21, was also reported in 3 studies [15,18,34] and has been implicated in retinal angiogenesis and inflammation in the diabetic context [38]. Additionally, miR-181c [15,25], miR-1179 [15,31] and miR-93 [16,18] were each reported in two studies, miR-93 has been associated with increased DR risk in T2DM [39].

## Clinical translation and the need for standardization

Our meta-analysis revealed considerable heterogeneity. One major source of variability stems from the type of biological matrix used. Although both plasma and serum were employed, plasma may offer a more reliable profile of circulating miRNAs [40]. Unlike serum, plasma avoids the confounding release of platelet-derived miRNAs during coagulation, which can distort expression profiles and lead to inconsistent results [41]. Another key issue is the lack of a universally accepted internal control for normalization. While U6 small nuclear RNA was the most commonly used reference gene across included studies, it is predominantly nuclear and may degrade in cell-free conditions such as plasma or serum, thus introducing bias [42]. Alternative reference miRNAs like miR-16-5p have demonstrated greater stability and may represent more appropriate normalization candidates in extracellular RNA research [43].

Other methodological factors, including the time elapsed between sample collection and processing [44], the type of miRNA extraction kit, and detection platform used, contribute further to between-study variability [45,46]. This lack of uniformity hinders the comparability of results and reduces their generalizability across different clinical contexts.

To overcome these barriers, the field urgently needs standardized pre-analytical protocols, consensus-based reporting guidelines, and the development of validated multi-miRNA diagnostic panels [47]. Furthermore, training and certification of technical personnel involved in miRNA handling and data interpretation would help minimize human error and increase reproducibility [48,49]. Without these improvements, the integration of miRNAs into clinical diagnostic workflows will remain theoretical, regardless of their promising statistical performance [50].

## Limitations

This meta-analysis has several limitations that must be considered when interpreting the findings. First, the protocol was not registered in Prospero. Second, a predominant number of the included studies were conducted in Chinese populations, which may introduce demographic bias and limit the generalizability of the findings to other ethnic groups, particularly Western cohorts. While this does not compromise internal validity, it underscores the need for broader geographic representation in future studies.

Third, the study designs were primarily case-control and cross-sectional, which are inherently more prone to bias than prospective cohort studies [51]. These designs may overestimate diagnostic accuracy due to spectrum bias or inappropriate patient selection [52]. Furthermore, randomized clinical trials are entirely lacking, reflecting both operational and methodological challenges in conducting such studies in this field.

Fourth, pre-analytical and analytical heterogeneity was substantial across studies. Variations in the biological sample type, normalization strategies, and miRNA isolation and detection platforms contributed significantly to inconsistency in results. Fifth, most studies did not report explicit diagnostic cut-off values for individual miRNAs. Although bivariate random-effects modeling allows for robust estimation of sensitivity, specificity, and AUC, the absence of threshold values limits the clinical applicability of the findings [53]. Diagnostic thresholds are essential for guiding real-world decision-making and should be established and validated in future research. The lack of cut-offs also reflects a broader issue: limited statistical training and methodological standardization among many investigators in the field [54].

## Conclusion

This meta-analysis demonstrates that circulating miRNAs exhibit promising diagnostic accuracy for distinguishing among various stages of DR, supporting their role as practical, non-invasive biomarkers. By stratifying the analysis into three clinically relevant comparison groups (CTL vs DR; T2DM vs DR; and NPDR vs PDR), we reduced inter-study heterogeneity

and generated more precise and clinically meaningful estimates of diagnostic performance. To further enhance their translational potential, miRNA expression profiling should be integrated with established clinical assessments and validated in well-designed prospective cohorts. We call upon the scientific and clinical community to establish international consensus on sample processing, normalization protocols, and diagnostic threshold definition. Only through rigorous standardization and prospective validation can circulating miRNAs be successfully incorporated into routine diagnostic workflows.

## Supporting information

**S1 File.** S1 Text. Complete search strategy. S1 Table. Prisma DTA abstract checklist. S2 Table. Prisma DTA checklist. S3 Table. Extraction data. S4 Table. Dataset used for STATA meta-analysis and meta-regression (CTL vs DR). S5 Table. Dataset used for STATA meta-analysis and meta-regression (T2DM vs DR). S6 Table. Dataset used for STATA meta-analysis and meta-regression (NPDR vs PDR).
(ZIP)

## Author contributions

**Conceptualization:** Miriam Martínez-Santos, Elías Martínez-López, Jorge M. Barcia.

**Data curation:** María Ybarra, Maria E. Pires, Chiara Ceresoni.

**Formal analysis:** Miriam Martínez-Santos, Elías Martínez-López.

**Funding acquisition:** Jorge M. Barcia, Maria Oltra, Javier Sancho-Pelluz.

**Investigation:** Miriam Martínez-Santos.

**Methodology:** Miriam Martínez-Santos, Elías Martínez-López, Maria Oltra.

**Project administration:** Jorge M. Barcia, Javier Sancho-Pelluz.

**Supervision:** Jorge M. Barcia, Maria Oltra, Javier Sancho-Pelluz.

**Visualization:** Miriam Martínez-Santos, María Ybarra.

**Writing – original draft:** Miriam Martínez-Santos, María Ybarra, Maria Oltra.

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
