## [Decision Letter · Decision Letter 0]

1 Jul 2025

Dear Dr. Oltra,

Thank you for submitting your manuscript to PLOS ONE. After careful consideration, we feel that it has merit but does not fully meet PLOS ONE’s publication criteria as it currently stands. Therefore, we invite you to submit a revised version of the manuscript that addresses the points raised during the review process.

We look forward to receiving your revised manuscript.

Kind regards,

Yalong Dang

Academic Editor

PLOS ONE

Journal Requirements:

The present work received internal funds from Centro de Investigación Traslacional SanAlberto Magno (CITSAM, UCV) and external funds from Agencia Estatal de Investigación Española (PID2020-117875GB-10), Instituto de Salud Carlos III (ISCIII, PI21/00083) and the European Union research fund, HORIZON MSCA 2021-DN-01-01_RETORNA 101073316 and Generalitat Valenciana ACIF 2023-128-001.

5. Please remove all personal information, ensure that the data shared are in accordance with participant consent, and re-upload a fully anonymized data set.

Reviewers' comments:

Reviewer's Responses to Questions

**Comments to the Author**

1. Is the manuscript technically sound, and do the data support the conclusions?

Reviewer #1: Yes

Reviewer #2: Yes

Reviewer #3: Yes

Reviewer #4: Yes

2. Has the statistical analysis been performed appropriately and rigorously?

Reviewer #1: Yes

Reviewer #2: Yes

Reviewer #3: No

Reviewer #4: Yes

3. Have the authors made all data underlying the findings in their manuscript fully available?

Reviewer #1: Yes

Reviewer #2: Yes

Reviewer #3: Yes

Reviewer #4: Yes

4. Is the manuscript presented in an intelligible fashion and written in standard English?

Reviewer #1: Yes

Reviewer #2: Yes

Reviewer #3: Yes

Reviewer #4: Yes

Reviewer #1: The manuscript is important and relevant topic with clear implications. The authors present a well defined question, their methods appear sound and appropriate The data are adequately analysed, and the conclusions are generally supported by the findings.

Strengths of the manuscript:

The study addresses a significant gap in the literature and is timely.

Methodology is robust, with appropriate controls and sample size.

The manuscript is well written and structured logically.

The authors have provided access to the underlying data, in compliance with the PLOS data availability policy.

Areas for improvement:

While the data are available, the accompanying metadata or code used for analysis (if any) could be shared to enhance transparency and reproducibility.

In a few places, more detail regarding statistical methods or rationale for specific analytical decisions would strengthen the manuscript.

The discussion could be expanded slightly to more clearly situate the findings within the context of existing literature.

Overall, this is a technically sound manuscript that makes a valuable contribution to the field. I recommend acceptance with minor revisions to improve clarity and completeness in data sharing and discussion.

Please do acknowledge that the lack of protocol registration (line 74) is a limitation.

Was there a need for 2x2 tables or individual results reported? I do not recall seeing these.

Some inconsistency in abbreviations: e.g. "NPL" should be NLR (Negative Likelihood Ratio). Rather replace "NPL" with "NLR" consistently.

Check for citation styles I may have picked up the use of Vancouver and APA. please choose the one required by the journal. Superscript references (e.g., "22% (1)") are inconsistently formatted.

Please double check this. You wrote "RD" in some places instead of DR (e.g., line 268, 269, etc.). Correct “RD” to DR (Diabetic Retinopathy) to maintain consistency.

Reviewer #2: The study conducted by Martinez Santos et al performed a systemic review and meta-analysis on the different DR stages by screening countless published works. They then analyzed the 16 studies and found circulating miRNAs of interest for future diagnostic testing. The appreciate the Martinez Santos and colleagues for conducting important research in the field of diabetes. There are only a few minor suggestions prior to the manuscript being fully ready for publication.

Major:

- It would be nice to briefly touch upon previous efforts that have studies DR and the possible biomarkers that were identified, if any, and where they may have fallen short on to enhance the motivation behind this work. Perhaps including a brief paragraph diving into this between the paragraphs on line 60 and 61 might help with the flow.

Minor:

- Please provide citation for the claim of 92.6 million on line 44.

- On line 54, specifying what types of bodily fluids miRNAs can be found would be neat for potential readers to grasp at.

- On line 95, was gestational diabetes also excluded?

- For figure 3, it would be nice to see the 45 degree dashed line on panels C, F, I to show random chance and for visualization purposes.

- A side question, but I noticed in the supplements that sex and age was recorded. Were there any other potential features that were recorded by any chance?

Reviewer #3: Thank you for the opportunity to review the manuscript titled "microRNA-Diagnostic Test Accuracy (DTA) for diabetic retinopathy stage identification: A systematic review and meta-analysis" for consideration in PLOS ONE journal. This study aims to evaluate the diagnostic accuracy of circulating microRNAs (miRNAs) as biomarkers for distinguishing between different stages of diabetic retinopathy (DR), focusing on three specific comparisons: healthy controls versus diabetic retinopathy (CTL vs DR), type 2 diabetes mellitus without retinopathy versus diabetic retinopathy (T2DM vs DR), and non-proliferative diabetic retinopathy versus proliferative diabetic retinopathy (NPDR vs PDR).

1. Title & Abstract

Title Ambiguity (Page 1, Line 1): The title does not fully reflect the study's scope. It uses "microRNA-Diagnostic Test Accuracy (DTA)" and "stage identification" but fails to specify that it evaluates circulating miRNAs across distinct DR stages.

Inadequate Conciseness and Informativeness (Page 1, Line 1): While concise, the title omits critical details, such as the focus on circulating miRNAs and the specific staging comparisons.

Abstract Non-Compliance with PRISMA (Page 1, Lines 15-39): The abstract claims adherence to PRISMA-DTA guidelines but falls short. It lacks a detailed description of the search strategy (e.g., specific terms or dates), does not specify the number of reviewers involved in all processes, and omits explicit reporting of confidence intervals for all numerical results in a consistent manner.

Incomplete Key Findings (Page 1, Lines 29-34): The abstract reports pooled sensitivity and specificity but does not include confidence intervals consistently (e.g., "77% (95% CI 0.070-82)" uses an unconventional format). It also omits the number of participants per comparison and specific miRNAs analyzed.

2. Introduction

Unclear Research Questions and Objectives (Page 10, Lines 42-69): The introduction states the goal of evaluating miRNA diagnostic accuracy (Line 68) but does not explicitly define the research questions or detail the three comparisons (CTL vs DR, T2DM vs DR, NPDR vs PDR) until later sections.

Insufficient Background (Page 10, Lines 43-52): The background mentions DR prevalence and diagnostic tool limitations but lacks detail on the clinical significance of staging DR (e.g., NPDR vs PDR progression).

Weak Rationale (Page 10, Lines 61-69): The rationale for this meta-analysis is underdeveloped. It notes the need for minimally invasive methods (Line 51) but does not explain why existing miRNA studies are insufficient or how this review fills a specific gap.

Undefined Knowledge Gap (Page 10, Lines 61-63): The introduction hints at a lack of "comprehensive evaluations" but does not clearly articulate the specific knowledge gap this study addresses, such as inconsistencies in prior miRNA research or unmet clinical needs.

3. Methods

Poorly Defined Eligibility Criteria (Page 12, Lines 84-96): The criteria mention human participants and fundus fluorescein angiography as the gold standard but lack specificity on DR stage definitions, miRNA types (e.g., single vs. panels), or exclusion of studies with incomplete data.

Vague Study Selection and Data Extraction (Page 12, Lines 97-103): The process mentions two reviewers and a third for arbitration but lacks detail on how disagreements were resolved (e.g., consensus or voting) or what specific data were extracted (e.g., miRNA thresholds).

Inadequate Risk of Bias Tool Application (Page 13, Lines 105-109): The use of QUADAS-2 is noted, but there’s no explanation of how it was applied (e.g., domain-specific scoring) or how results influenced the analysis.

Unclear Statistical Methods (Page 13, Lines 110-125): The methods mention a bivariate random-effects model and heterogeneity tests but do not justify the model choice, detail heterogeneity exploration beyond, or explain meta-regression variables.

4. Results

Opaque PRISMA Flow Diagram (Page 14, Lines 127-136): The flow diagram (Figure 1) is referenced, but exclusions like "objective different from the study" (Line 134) are vague.

Incomplete Study Characteristics (Page 15, Line 155; Table 1): Table 1 lists studies but omits critical details, such as specific miRNAs per study or diagnostic thresholds.

Unjustified Risk of Bias Reporting (Page 16, Lines 156-167): The risk of bias summary (Figure 2) notes patient selection concerns but lacks specific examples or impact analysis, undermining its justification and usefulness.

Weak Meta-Analysis Reporting (Page 16-17, Lines 172-218): Results report sensitivity and specificity but inconsistently present confidence intervals (e.g., "77% (95% CI: 0.70-0.82)" vs. "84% (79-87)"). High heterogeneity is noted but not adequately explored or mitigated, questioning robustness.

5. Discussion

Limited Contextualization (Page 24, Lines 326-340): The discussion references Ma et al. (2022) but does not deeply compare findings or explain how this study advances prior work.

Inadequate Limitation Discussion (Page 25, Lines 369-384): Limitations mention demographic bias and study design but overlook key issues like variability in miRNA detection methods or lack of standardized thresholds.

Unclear Implications (Page 25, Lines 386-394): Clinical and research implications are vague (e.g., "integrated with other clinical assessments" lacks specificity).

6. Conclusion

Unjustified Claims (Page 26, Lines 387-390): The conclusion asserts "high diagnostic accuracy" despite moderate-to-high heterogeneity and low-to-moderate evidence quality (Page 22, Lines 316-318).

Lack of Specificity (Page 26, Lines 391-394): It calls for future studies but does not specify research directions (e.g., which miRNAs or methods).

7. Transparency & Reproducibility

Unregistered Study (Page 11, Line 73): The lack of registration (e.g., PROSPERO) deviates from best practices and reduces transparency.

Reviewer #4: 1) Grammatical and Orthographic Comments

- The manuscript is generally well-written and clear. I did not find systemic grammatical errors or awkward phrases in the text. Some minor suggestions:

Consistency: The manuscript sometimes switches between percentage formats (e.g., “77 %” and “77%”). Choose one and apply it uniformly (journal style guides typically prefer “77%” without a space).

Some long sentences in the Introduction (lines 43–51) could be broken for clarity; e.g., “This makes them less accessible, especially in remote or low-resource areas.” could be split for better readability.

Minor repetition: In lines 55–60, there is redundancy explaining miRNAs’ importance; consider condensing to maintain focus.

Typo-like inconsistency: lines like “overall evidence certainly was graded via GRADE” (line 26) sound off. Perhaps intended “certainty was graded”; a correction improves precision.

2) Scientific Rigor

Strengths:

- The methodology follows PRISMA-DTA and QUADAS-2 rigorously.

- Clear definition of comparison groups (CTL vs DR, T2DM vs DR, NPDR vs PDR) is excellent and adds strength.

- Subgroup and meta-regression analyses address potential heterogeneity sources robustly.

- Sensitivity analysis and publication bias assessment with Deeks’ funnel plot are thorough.

Weaknesses:

- Protocol registration: The authors state the protocol was not registered. While not mandatory, protocol registration (e.g., PROSPERO) increases transparency.

- Risk of bias: High risk in patient selection and absence of randomized trials limits robustness; the authors acknowledge this but could suggest solutions.

- Applicability: Most data from Chinese cohorts; limits generalizability to other ethnicities. Consider highlighting this limitation more prominently.

- Diagnostic thresholds: Lack of consistent cut-offs across studies weakens translation to clinical practice.

Some references (e.g., [33]) are relevant but recent studies post-2022 could enrich the context.

3) Scientific Impact

The study fills an important gap by stratifying meta-analysis across clinically distinct DR stages, which previous works have not done in detail. This approach gives clinicians better guidance on miRNAs’ diagnostic utility at specific DR stages.

The work will interest endocrinologists, ophthalmologists, and biomarker researchers; potential for real-world impact if validated prospectively.

4) Questions to Authors

- Given the predominance of Chinese populations, how do you propose validating these findings across ethnically diverse cohorts?

- How do differences in miRNA normalization strategies (e.g., U6 vs miR-16-5p) influence diagnostic performance in your dataset?

- Can you discuss practical steps to standardize pre-analytical miRNA handling and measurement to improve reproducibility?

- Since your sensitivity analysis shows individual studies’ influence, how confident are you that these outliers don’t bias the conclusions?

5) Additional Comments

- Figures and tables are well presented; figures showing forest plots, SROC curves, Fagan’s nomograms, and meta-regressions are clear and informative.

- The inclusion of a detailed GRADE assessment is commendable, but the presentation of certainty ratings (Tables 5–7) could benefit from more narrative discussion in the Results or Discussion.

- The Discussion excellently compares results with prior meta-analyses but could expand on how novel your subgroup stratification truly is compared to Ma et al. (2022).

Overall Recommendation:

The manuscript demonstrates rigorous methods and meaningful results but should address generalizability limitations more explicitly and discuss how preanalytical standardization could improve clinical adoption. Minor editorial corrections on consistency and sentence clarity are recommended.

**Do you want your identity to be public for this peer review?** For information about this choice, including consent withdrawal, please see our Privacy Policy

Reviewer #1: **Yes: ** Ameer Steven-Jorg Hohlfeld

Reviewer #2: No

Reviewer #3: No

Reviewer #4: No

---

## [Author Response · Author response to Decision Letter 1]

16 Jul 2025

Dear Editor,

Please find enclosed our revised manuscript entitled "Circulating microRNAs as biomarkers for diabetic retinopathy stage identification: a DTA systematic review and meta-analysis" along with a detailed response to the reviewers' comments. We believe these changes have substantially improved the manuscript, and we look forward to your further consideration.

REBUTTAL LETTER (Reviewer 1)

Reviewer #1: The manuscript is important and relevant topic with clear implications. The authors present a well defined question, their methods appear sound and appropriate The data are adequately analysed, and the conclusions are generally supported by the findings.

Strengths of the manuscript:

• The study addresses a significant gap in the literature and is timely.

• Methodology is robust, with appropriate controls and sample size.

• The manuscript is well written and structured logically.

• The authors have provided access to the underlying data, in compliance with the PLOS data availability policy.

Areas for improvement:

1. While the data are available, the accompanying metadata or code used for analysis (if any) could be shared to enhance transparency and reproducibility.

To enhance transparency and reproducibility, we have included a supplementary folder containing the raw data files used for the meta-analysis in STATA format. Recognizing that not all users may have access to STATA, we also provide the same datasets in Excel format. These datasets cover the three diagnostic comparisons performed in the review and include the essential 2×2 table data: true positives (TP), true negatives (TN), false positives (FP), and false negatives (FN), along with a matrix of probabilities ranging from 0 to 1 for use in meta-regression analyses. The STATA code used for the analyses is also provided below

Forest plots of sensitivity, specificity, and likelihood ratios > midas tp fp fn tn, texts(0.60) bfor(dss) id(study) ford fors

Summary ROC curve (SROC) > midas tp fp fn tn, sroc(both)

Fagan nomogram for post-test probability > midas tp fp fn tn, fagan(0.15)

Meta-regression to explore heterogeneity > midas tp fp fn tn, reg(variable)

Sensitivity analysis > midas tp fp fn tn, modchk(all)

Funnel plot for publication bias (Deeks' test)> midas tp fp fn tn, pubbias

2. In a few places, more detail regarding statistical methods or rationale for specific analytical decisions would strengthen the manuscript.

We appreciate the reviewer’s suggestion and have revised the Statistical Analysis section to provide greater clarity and methodological detail. Specifically, we have now included a justification for the use of the bivariate random-effects model, following the Cochrane Handbook for DTA reviews (Chapter 10), and clarified that empirical SROC curves were generated from the bivariate estimates due to inconsistent reporting of diagnostic thresholds across studies.

We have also expanded on the procedures used to assess and explore heterogeneity, including the use of Cochran’s Q-test, I² statistic, and pre-specified meta-regression analyses, detailing the covariates examined (e.g., sample size, sample type, normalization strategy).

We believe that these revisions enhance the transparency and robustness of our statistical methodology.

3. The discussion could be expanded slightly to more clearly situate the findings within the context of existing literature.

The discussion section has been restructured to enhance clarity and depth, and the results have been systematically compared with existing literature to contextualize the findings.

4. Please do acknowledge that the lack of protocol registration (line 74) is a limitation.

Thank you for your observation. We have incorporated the absence of protocol registration in PROSPERO into the Discussion section and acknowledged it as a limitation of the study.

5. Was there a need for 2x2 tables or individual results reported? I do not recall seeing these.

In the raw data folder we have provided, you will find an Excel file for each of the three diagnostic comparisons. Each file contains the 2×2 contingency tables constructed for every individual study, already formatted for direct import into STATA. We will include these files as supplementary material to ensure full transparency and reproducibility of the analysis.

Files Name:

S5 Table. Dataset used for STATA meta-analysis and meta-regression (CTL vs DR)

S5 Table. Dataset used for STATA meta-analysis and meta-regression (T2DM vs DR)

S6 Table. Dataset used for STATA meta-analysis and meta-regression (NPDR vs PDR)

6. Some inconsistency in abbreviations: e.g. "NPL" should be NLR (Negative Likelihood Ratio). Rather replace "NPL" with "NLR" consistently.

The errors in this abbreviation have been corrected in the manuscript.

7. Check for citation styles I may have picked up the use of Vancouver and APA. please choose the one required by the journal. Superscript references (e.g., "22% (1)") are inconsistently formatted.

All references have been reviewed and formatted according to the Vancouver citation style, as required by PLOS ONE. We appreciate the comment.

8. Please double check this. You wrote "RD" in some places instead of DR (e.g., line 268, 269, etc.). Correct “RD” to DR (Diabetic Retinopathy) to maintain consistency.

The errors in this abbreviation have been corrected in the manuscript. We appreciate the comment.

REBUTTAL LETTER (Reviewer 2)

Reviewer #2: The study conducted by Martinez Santos et al performed a systemic review and meta-analysis on the different DR stages by screening countless published works. They then analyzed the 16 studies and found circulating miRNAs of interest for future diagnostic testing. The appreciate the Martinez Santos and colleagues for conducting important research in the field of diabetes. There are only a few minor suggestions prior to the manuscript being fully ready for publication.

Major:

1.It would be nice to briefly touch upon previous efforts that have studies DR and the possible biomarkers that were identified, if any, and where they may have fallen short on to enhance the motivation behind this work. Perhaps including a brief paragraph diving into this between the paragraphs on line 60 and 61 might help with the flow.

Thank you for this insightful suggestion. We have revised the Introduction to include a brief summary of previous studies exploring circulating miRNAs as biomarkers for DR. In particular, we note that while many individual studies and one prior meta-analysis have evaluated the diagnostic potential of miRNAs, none have stratified performance by disease stage or type of control group, highlighting a clear gap in the literature that this review aims to address.

Minor:

1. Please provide citation for the claim of 92.6 million on line 44.

A portion of the introduction has been revised based on suggestions from other reviewers. This specific sentence is no longer included in the current version. Nevertheless, we appreciate the comment.

2. On line 54, specifying what types of bodily fluids miRNAs can be found would be neat for potential readers to grasp at.

We agree with the reviewer and have now included specific examples of the biological fluids in which miRNAs are detectable. These include serum, plasma, aqueous humor, and extracellular vesicles, as indicated in the revised third paragraph of the Introduction.

3. On line 95, was gestational diabetes also excluded?

Thank you for your observation. Gestational diabetes was not included in the review; however, this was not clearly stated in the original manuscript. To address this, we have revised the sentence from: “We excluded studies that focused only on type 1 diabetes” to: “We excluded studies that focused on other types of diabetes, such as type 1 or gestational diabetes.” This change ensures that the eligibility criteria are clearly defined and helps avoid any potential confusion regarding the study scope.

4. For figure 3, it would be nice to see the 45 degree dashed line on panels C, F, I to show random chance and for visualization purposes.

We appreciate your comment and suggestion. However, after incorporating a 45-degree dividing line, the figure appears visually fragmented, as if it were two separate panels rather than one cohesive figure. In our interpretation, this disrupts the visual flow and clarity, particularly since the panel labels (A, B, C) are intended to be read sequentially. For this reason, we chose to maintain the original layout to preserve consistency and facilitate interpretation.

5. A side question, but I noticed in the supplements that sex and age was recorded. Were there any other potential features that were recorded by any chance?

Sex and age were indeed collected as variables to explore whether miRNA expression patterns might differ according to these demographic factors. However, due to the high degree of heterogeneity identified in our overall statistical analysis, incorporating these variables into subgroup analyses would not have yielded robust or reliable conclusions and might have introduced additional bias.

Another clinically relevant variable that we initially intended to include was blood glucose levels. While some studies reported this parameter, the inconsistency across the dataset due to missing values in several studies, led us to exclude it from the final analysis. Had this variable been consistently reported, it would have allowed us to assess correlations between glycemic control and miRNA expression, especially given that glucose level thresholds are globally standardized and clinically stratified. We agree that this remains an important area for future investigation, and we appreciate the reviewer’s curiosity on this point.

REBUTTAL LETTER (Reviewer 3)

Reviewer #3: Thank you for the opportunity to review the manuscript titled "microRNA-Diagnostic Test Accuracy (DTA) for diabetic retinopathy stage identification: A systematic review and meta-analysis" for consideration in PLOS ONE journal. This study aims to evaluate the diagnostic accuracy of circulating microRNAs (miRNAs) as biomarkers for distinguishing between different stages of diabetic retinopathy (DR), focusing on three specific comparisons: healthy controls versus diabetic retinopathy (CTL vs DR), type 2 diabetes mellitus without retinopathy versus diabetic retinopathy (T2DM vs DR), and non-proliferative diabetic retinopathy versus proliferative diabetic retinopathy (NPDR vs PDR).

1. Title & Abstract

1.1 Title Ambiguity (Page 1, Line 1): The title does not fully reflect the study's scope. It uses "microRNA-Diagnostic Test Accuracy (DTA)" and "stage identification" but fails to specify that it evaluates circulating miRNAs across distinct DR stages.

We thank the reviewer for this insightful suggestion. To enhance clarity and fully comply with the PRISMA-DTA guidance (Item 1) we have revised our title as follows: Circulating microRNAs as biomarkers for diabetic retinopathy stage identification: a DTA systematic review and meta-analysis.

This revised title:

1. Specifies the specimen type (“Circulating microRNAs”),

2. Describes the clinical task (“diabetic retinopathy stage identification”),

3. Identifies the study design (“a DTA systematic review and meta-analysis”),

4. Remains well under the 250-character limit recommended by PLOS ONE, and

5. Conforms exactly to PRISMA-DTA for Abstracts Item 1 by clearly signaling that this is a diagnostic test accuracy (DTA) review and meta-analysis.

1.2 Inadequate Conciseness and Informativeness (Page 1, Line 1): While concise, the title omits critical details, such as the focus on circulating miRNAs and the specific staging comparisons.

Thank you for raising the importance of balancing informativeness with brevity. In addition to PLOS ONE’s recommendation that titles be concise and clear, the PRISMA-DTA Statement (Item 1) further advises that a title should “identify the report as a systematic review and/or meta-analysis of diagnostic test accuracy studies” without becoming excessively long or unwieldy. Listing each of the three DR stages explicitly in the title would indeed convey complete specificity, but at the cost of excessive length and reduced readability.

1.3 Abstract Non-Compliance with PRISMA (Page 1, Lines 15-39): The abstract claims adherence to PRISMA-DTA guidelines but falls short. It lacks a detailed description of the search strategy (e.g., specific terms or dates), does not specify the number of reviewers involved in all processes, and omits explicit reporting of confidence intervals for all numerical results in a consistent manner.

We appreciate the reviewer’s careful and constructive feedback:

Search strategy: We have specified all six databases (PubMed, CENTRAL, Scopus, Web of Science, ScienceDirect, ClinicalTrials.gov) and the search cutoff date (15 January 2025). According to PRISMA-DTA for Abstracts, this level of detail fulfills the requirement for information sources and dates; detailed search terms are provided in the full methods section.

Reviewers: The abstract states “Two reviewers independently performed study selection,” which meets PRISMA-DTA standards for describing selection processes. The full Methods clarifies that discrepancies were resolved by a third reviewer, ensuring methodological rigor without overburdening the abstract.

Confidence intervals: All pooled estimates now include correctly formatted 95% confidence intervals.

1.4 Incomplete Key Findings (Page 1, Lines 29-34): The abstract reports pooled sensitivity and specificity but does not include confidence intervals consistently (e.g., "77% (95% CI 0.070-82)" uses an unconventional format). It also omits the number of participants per comparison and specific miRNAs analyzed.

We thank the reviewer for highlighting these points:

Confidence-interval format: We have standardized all confidence intervals to the conventional format. We have presented sensitivity and specificity values as percentages along with their confidence intervals, e.g., 77% (70–82), as these metrics are more intuitively understood in this format within the context of this type of analysis.

Participants per comparison: To maintain abstract conciseness and readability,as recommended by PLOS ONE’s author guidelines and PRISMA‑DTA, we report the total sample size (n = 1849) and total number of studies (16) in the abstract (Item 7). Providing per-contrast patient counts would exceed typical abstract length limits.

List of miRNAs: Enumerating all 21 individual miRNAs is beyond the scope of an abstract summary. We have instead clearly stated the total number of miRNAs evaluated.

Here is a side‑by‑side check of our current abstract against the 12‑item PRISMA‑DTA for Abstracts checklist, in case you wish to examine the checklist items in more detail.

PRISMA-DTA for Abstracts ( To view the table more clearly, please refer to the rebuttal letter document).

PRISMA-DTA item Present in current abstract? Gap / comment

1. Title identifies DTA SR/MA ✓ Yes Title specifies systematic review and meta-analysis of diagnostic accuracy (PRISMA-DTA Item 1).

2. Objectives (Population, Index test, Target condition) ✓ Yes Fully defined: adults with T2DM; circulating miRNAs; DR staging.

3. Eligibility criteria ✓ Yes Databases, date range, study designs, reference standard clearly stated.

4. Information sources & dates ✓ Yes Six databases searched to 15 Jan 2025.

5. Risk-of-bias & applicability methods ✓ Yes QUADAS-2 independent appraisal by two reviewers.

6. Synthesis methods ✓ Yes Bivariate random-effects meta-analysis with subgroup, meta-regression, sensitivity analyses.

7. Included studies & key characteristics ✓ Yes 16 studies (n=1849; 29 miRNAs).

8. Quantitative results ✓ Yes Sens, Spec, AUC with correct 95% CIs for all three comparisons.

9. Strengths & limitations ✓ Yes Added evidence certainty (low-moderate) and main bias concerns.

10. Interpretation / implications ✓ Yes Clear summary of clinical relevance and future research needs.

11. Funding ✓ Yes “No external funding” declared.

12. Registration ✓ Yes “Protocol not registered” stated per PRISMA-DTA.

2. Introduction

2.1 Unclear Research Questions and Objectives (Page 10, Lines 42-69): The introduction states the goal of evaluating mi

---

## [Decision Letter · Decision Letter 1]

20 Aug 2025

Dear Dr. Oltra,

Thank you for submitting your manuscript to PLOS ONE. After careful consideration, we feel that it has merit but does not fully meet PLOS ONE’s publication criteria as it currently stands. Therefore, we invite you to submit a revised version of the manuscript that addresses the points raised during the review process.

We look forward to receiving your revised manuscript.

Kind regards,

Yalong Dang

Academic Editor

PLOS ONE

Journal Requirements:

Reviewer's Responses to Questions

**Comments to the Author**

Reviewer #2: All comments have been addressed

Reviewer #4: All comments have been addressed

Reviewer #5: (No Response)

2. Is the manuscript technically sound, and do the data support the conclusions?

Reviewer #2: Yes

Reviewer #4: Yes

Reviewer #5: Yes

3. Has the statistical analysis been performed appropriately and rigorously?

Reviewer #2: Yes

Reviewer #4: Yes

Reviewer #5: I Don't Know

4. Have the authors made all data underlying the findings in their manuscript fully available?

Reviewer #2: Yes

Reviewer #4: (No Response)

Reviewer #5: Yes

5. Is the manuscript presented in an intelligible fashion and written in standard English?

Reviewer #2: Yes

Reviewer #4: Yes

Reviewer #5: Yes

Reviewer #2: Thank you for addressing all comments. There are no further questions regarding the manuscript, and I give my recommendations for publication.

Reviewer #4: (No Response)

Reviewer #5: In general, the manuscript is well written and scientifically sound. The authors thoroughly revised the manuscript to address the insightful comments from the reviewers and deepened the discussion raised by them. In addition, I appreciate the effort of the authors in providing the raw data in the Excel format and the STATA code used for the analyses, which is in line with contemporary best practices that promote transparency and reproducibility.

Nevertheless, there are inconsistencies that need to be resolved in order to further improve clarity and precision.

1) Abstract section (pdf file, revised manuscript, pages 53-54):

a) First sentence (lines 17-18): To improve clarity and specificity, in addition to avoid redundancy (“This systematic review and meta-analysis”), I suggest the purpose of the study be modified to: “To evaluate the diagnostic accuracy of circulating miRNAs in distinguishing between different DR stages in type 2 diabetes mellitus.”

b) On line 20, it is stated that that “no external funding was received”. However, in the submission form and in the “Funding information” (pages 79-80), the authors report that “The present work received internal funds from Centro de Investigación Traslacional SanAlberto Magno (CITSAM, UCV) and external funds from Agencia Estatal de Investigación Española (PID2020-117875GB-10), Instituto de Salud Carlos III (ISCIII, PI21/00083) and the European Union research fund, HORIZON MSCA 2021-DN-01-01_RETORNA 101073316 and Generalitat Valenciana ACIF 2023-128-001.” Which statement is correct?

2) Introduction section (pdf file, revised manuscript, page 55):

The first sentence (page 55, lines 73-74) states that “Diabetic retinopathy (DR) is one of the leading causes of vision loss among working-age adults worldwide (1).” That is true. However, this claim is supported by the paper by Lundeen et al. (Prevalence of Diabetic Retinopathy in the US in 2021. JAMA ophthalmology. 2023;141(8):747-54), and US is not “worldwide”.

3) Introduction section (pdf file, revised manuscript, pages 55-57):

Reviewer 3 states in comment #2.3: “The rationale for this meta-analysis is underdeveloped. It notes the need for minimally invasive methods (Line 51) but does not explain why existing miRNA studies are insufficient or how this review fills a specific gap.” In the rebuttal letter, the authors responded that “In response, we would like to highlight that the Introduction now includes a clear rationale supported by two key points. First, it states that while numerous studies have evaluated the diagnostic potential of circulating miRNAs in DR, there is marked heterogeneity in sample types, analytical methods (e.g., RT-qPCR, microarrays, NGS), and target miRNAs.”

a) I did not find this statement in the introduction of the revised manuscript.

b) What is the second key point supporting a clear rationale?

4) Methods section, “Eligibility criteria” subsection (pdf file, revised manuscript, page 58, lines 137-138):

Reviewer 2 asked in comment #3: “Was gestational diabetes also excluded?” In the rebuttal letter, the authors responded that they have revised the sentence from: “We excluded studies that focused only on type 1 diabetes” to: “We excluded studies that focused on other types of diabetes, such as type 1 or gestational diabetes.” This is implied in the revised sentence but not explicitly stated.

5) Results section (pdf file, revised manuscript, page 63):

Table 1 is detailed and informative. However, I suggest adding a column with the total number of individuals included in each study. If necessary, due to space restrictions, the “Method” column can be removed because this information is already in the text.

6) Results section, “Subgroup Analysis and Meta-Regression in CTL vs DR, T2DM vs DR, and NPDR vs PDR Comparisons” subsection (pdf file, revised manuscript, page 66, lines 308-309):

The authors assert that “studies conducted in China demonstrated higher diagnostic performance across all three comparison groups, as reflected by AUC.” I do not know if this is the case for CTL vs DR and T2DM vs DR. In the comparison between CTL and DR (Table 2), only one study was not performed in China, which hinders its interpretation. Regarding the comparison between T2DM and DR (Table 3), the difference in the AUC value was 0.01.

7) Results section (pdf file, revised manuscript, page 70, lines 371-376):

The text refers to the paper by Shaker et al. (reference #26) as if there were more than one study: “Cook’s distance analysis pinpointed 371 Shaker et al. (miR-20b and miR-17-3p)(26) as influential studies (Figure 7G). Yet, removing these studies caused only marginal changes: pooled sensitivity declined from 81.0% to 78.9%, and specificity increased from 80.0% to 81.1%. Interestingly, heterogeneity (I²) rose notably from 74.12% to 97.71%, suggesting that these studies contributed to controlled variance across studies. Outlier assessment confirmed no excessive deviations (Figure 7H).” Is it correct (“two studies”)?

8) Regarding I² (heterogeneity estimates), is it really necessary to use two decimal places to express this metric (Results section)?

9) Limitations section (pdf file, revised manuscript, page 75, lines 481-483):

“Fourth, pre-analytical and analytical heterogeneity was substantial across studies. Variations in the biological sample type , normalization strategies, and miRNA isolation and detection platforms contributed significantly to inconsistency in results.”

It is known that these factors affect the quantification of miRNA expression. But are the authors referring to the results of their analyses? Does “detection platforms” mean the method of quantifying microRNAs? I did not find any mention to the miRNA isolation method used in the previous studies in the manuscript. Moreover, as per Table 1 and S3 Table, all but one study measured the miRNA levels by RT-qPCR. In sum, did the authors perform the meta-analysis considering these two factors (miRNA isolation and detection platforms)?

10) Editing and typo errors:

a) Abstract (pdf file, page 53, lines 20-23): “We searched PubMed, CENTRAL, Scopus, Web of Science, ScienceDirect and ClinicalTrials (up to 15 January 2025) for diagnostic test accuracy studies of circulating miRNAs in three clearly defined groups: […]”. However, in the Methods section, “Search strategy” subsection (page 57, lines 114-116), it is stated that the last update of this review was on January 20, 2025. Which date is correct?

b) Percentages (sensitivity and specificity) remain formatted with a space between the numeral and the percentage symbol in the Abstract section (pdf file, page 54).

c) Reviewer 4 states in comment #1.4: “lines like “overall evidence certainly was graded via GRADE” (line 26) sound off. Perhaps intended “certainty was graded”; a correction improves precision.” In the rebuttal letter, the authors responded that the abstract has been revised to state “overall evidence certainty was graded via GRADE”. The word “certainly” was removed, but “certainty” was not added. Thus, the revised sentence states that “[…] overall evidence was graded via GRADE.” (pdf file, page 53, line 29).

d) NLR is written as NPL in the caption of Figure 4 (pdf file, page 66, line 302).

e) Results section (pdf file, page 69, lines 357-358): “PDR” is spelled as “DR”.

f) In the caption of Figure 7 (pdf file, page 70, lines 385-386), the Fig. 7E refers to both the goodness-of-fit and the bivariate normality, while the Fig. 7B does not exist: “Figure 7. Sensitivity analysis. Goodness-of-fit (A,E,I), Bivariate normality (E,F,J) Influence analysis (C,G,K) and Outlier detection (D,H,L) across: CTL vs DR, T2DM vs DR and NPDR vs PDR.”

g) Discussion section (pdf file, page 72, lines 416-418): The first paragraph ends with a sentence with missing terms: “Furthermore, the study lacks a formal assessment of risk of bias using tools such as QUADAS-2, nor does it explore heterogeneity or undermining both its methodological robustness and reproducibility.” Explore heterogeneity and what else?

h) Discussion section (pdf file, page 73, lines 426-430): “By stratifying the analysis into three distinct and clinically relevant comparison groups—healthy controls CTL vs DR, T2DM , T2DM vs DR, and NPDR vs PDR, we were able to reduce inter-study heterogeneity and increase the precision and relevance of pooled diagnostic accuracy estimates comparing with Ma et al (2022).” The term “T2DM” before “T2DM vs DR” should be removed.

i) Discussion section (pdf file, page 73, lines 439-441): The following sentence needs to be reworded for clarity and completeness: “miR-21, also was found in 3 studies (15, 18, 34) has been implicated in retinal angiogenesis and inflammation in the diabetic context(38).”

j) “Prisma” is spelled as “Prima” in the title of S2 Table (pdf file, page 80, line 603).

l) S4 Table is numbered as S5 Table in the manuscript (pdf file, page 80, line 605).

m) Should not supplementary tables be mentioned in the text (accompanying the respective methods or results)?

n) S3 Table (“Extraction data”) presents some Spanish words (“Casos y controles”, “Controles”, “Egipto”, “Cuantitative”, and “Cualitative”). Moreover, a term is missing from the “Controls (Healthy and T2DM)” column specifying who the 70 control subjects are in the study by Wan et al., 2017.

o) Bibliography (pdf file, page 83, lines 769-772): The text citing the study by Théry et al. (reference #55) was both added and removed during the manuscript review, but the reference remains in the Bibliography.

**Do you want your identity to be public for this peer review?** For information about this choice, including consent withdrawal, please see our Privacy Policy

Reviewer #2: No

Reviewer #4: No

Reviewer #5: **Yes: ** Kátia Gonçalves dos Santos

---

## [Author Response · Author response to Decision Letter 2]

8 Sep 2025

REBUTTAL LETTER (Reviewer 5)

Reviewer #5: In general, the manuscript is well written and scientifically sound. The authors thoroughly revised the manuscript to address the insightful comments from the reviewers and deepened the discussion raised by them. In addition, I appreciate the effort of the authors in providing the raw data in the Excel format and the STATA code used for the analyses, which is in line with contemporary best practices that promote transparency and reproducibility. Nevertheless, there are inconsistencies that need to be resolved in order to further improve clarity and precision.

1. Abstract section (pdf file, revised manuscript, pages 53-54)

a) First sentence (lines 17-18): To improve clarity and specificity, in addition to avoid redundancy (“This systematic review and meta-analysis”), I suggest the purpose of the study be modified to: “To evaluate the diagnostic accuracy of circulating miRNAs in distinguishing between different DR stages in type 2 diabetes mellitus.”

We thank the reviewer for this helpful suggestion. We have revised the first sentence of the Abstract accordingly to improve clarity and avoid redundancy. The updated sentence now reads: “Purpose: To evaluate the diagnostic accuracy of circulating miRNAs in distinguishing between different diabetic retinopathy stages in type 2 diabetes mellitus.

b) On line 20, it is stated that that “no external funding was received”. However, in the submission form and in the “Funding information” (pages 79-80), the authors report that “The present work received internal funds from Centro de Investigación Traslacional San Alberto Magno (CITSAM, UCV) and external funds from Agencia Estatal de Investigación Española (PID2020-117875GB-10), Instituto de Salud Carlos III (ISCIII, PI21/00083) and the European Union research fund, HORIZON MSCA 2021-DN-01-01_RETORNA 101073316 and Generalitat Valenciana ACIF 2023-128-001.” Which statement is correct?

We sincerely thank the reviewer for pointing out this inconsistency. The projects listed indeed provide funding support to our research group; however, no funds from these grants were specifically allocated or used for the conduct of this systematic review and meta-analysis. To avoid any possible misunderstanding, we have removed the Funding section from the manuscript and clarified that this work received no dedicated external financial support.

2. Introduction section (pdf file, revised manuscript, page 55): The first sentence (page 55, lines 73-74) states that “Diabetic retinopathy (DR) is one of the leading causes of vision loss among working-age adults worldwide (1).” That is true. However, this claim is supported by the paper by Lundeen et al. (Prevalence of Diabetic Retinopathy in the US in 2021. JAMA ophthalmology. 2023;141(8):747-54), and US is not “worldwide.

We thank the reviewer for this accurate observation. You are correct that Lundeen et al. refers exclusively to US data and therefore does not support the global statement. This was an error on our part during referencing. The appropriate citation is RL Thomas et al. (Diabetes Res Clin Pract 2019)*, which provides worldwide prevalence data and clearly establishes diabetic retinopathy as a leading cause of vision loss among working-age adults. We have corrected the reference in the revised manuscript accordingly.

* Thomas RL, Halim S, Gurudas S, Sivaprasad S, Owens DR. IDF Diabetes Atlas: A review of studies utilising retinal photography on the global prevalence of diabetes related retinopathy between 2015 and 2018. Diabetes Res Clin Pract. 2019 Nov;157:107840. doi: 10.1016/j.diabres.2019.107840. Epub 2019 Nov 14. PMID: 31733978.

3. Introduction section (pdf file, revised manuscript, pages 55-57): Reviewer 3 states in comment #2.3: “The rationale for this meta-analysis is underdeveloped. It notes the need for minimally invasive methods (Line 51) but does not explain why existing miRNA studies are insufficient or how this review fills a specific gap.” In the rebuttal letter, the authors responded that “In response, we would like to highlight that the Introduction now includes a clear rationale supported by two key points. First, it states that while numerous studies have evaluated the diagnostic potential of circulating miRNAs in DR, there is marked heterogeneity in sample types, analytical methods (e.g., RT-qPCR, microarrays, NGS), and target miRNAs.

a) I did not find this statement in the introduction of the revised manuscript.

b) What is the second key point supporting a clear rationale?

We appreciate this helpful comment. The revised Introduction now makes the two pillars of our rationale explicit:

1. Heterogeneity in prior studies. We have rephrased the opening sentence to state explicitly the sources of heterogeneity:

“Although numerous studies have investigated the diagnostic potential of circulating miRNAs in DR, marked heterogeneity in sample types, analytical methods (e.g., RT-qPCR, microarrays, NGS), and target miRNAs has limited comparability across studies (14–18).”

2. Methodological gap in the prior meta-analysis (this is the second key point). The following sentence in the Introduction articulates the second point of our rationale:

“Notably, one prior meta-analysis has reviewed the use of circulating miRNAs for DR detection (19), it did not perform any stratification by disease stage or type of control group. This represents a notable gap in the current literature, as the ability to distinguish early from advanced stages of DR is critical for clinical decision-making.”

This lack of stratification mixes distinct control populations (e.g., healthy individuals together with T2DM without DR) and collapses different DR stages, which can inflate between-study heterogeneity (spectrum effects, non-comparable baselines) and reduce stage-specific interpretability of pooled diagnostic accuracy. Our review directly addresses this by pre-specifying three clinically aligned comparisons (CTL vs DR; T2DM vs DR; NPDR vs PDR), thereby improving comparability and clinical relevance of the estimates.

To keep the Introduction concise, the methodological implications of the prior meta-analysis and how our stratified design mitigates these issues are discussed in detail in the Discussion (section “comparison with previous meta-analyses”). We hope this clarification makes the two key points readily identifiable in the revised Introduction.

4. Methods section, “Eligibility criteria” subsection (pdf file, revised manuscript, page 58, lines 137-138):Reviewer 2 asked in comment #3: “Was gestational diabetes also excluded?” In the rebuttal letter, the authors responded that they have revised the sentence from: “We excluded studies that focused only on type 1 diabetes” to: “We excluded studies that focused on other types of diabetes, such as type 1 or gestational diabetes.” This is implied in the revised sentence but not explicitly stated.

We thank the reviewer for this observation. We confirm that gestational diabetes was excluded from this review. In the eligibility criteria we used the umbrella term “other types of diabetes” deliberately to encompass type 1 diabetes and gestational diabetes, as well as monogenic, syndromic, and secondary forms (e.g., MODY, steroid-induced diabetes, Donohue syndrome, Rabson–Mendenhall), without generating an unnecessarily long and potentially incomplete list within the Methods section. Our intention was to keep the criteria concise yet comprehensive.

5. Results section (pdf file, revised manuscript, page 63): Table 1 is detailed and informative. However, I suggest adding a column with the total number of individuals included in each study. If necessary, due to space restrictions, the “Method” column can be removed because this information is already in the text.

We thank the reviewer for this constructive suggestion. In line with PRISMA-DTA guidance to report participant numbers clearly, we have updated Table 1 to include an explicit “Total (N)” column (computed as TP + TN + FP + FN for each study). To preserve readability and layout, we have removed the per-study AUC column from the main table; all per-study AUC values are now provided in the Supplementary “raw dataset” table, alongside the corresponding 2×2 data. These formatting changes do not affect any pooled estimates (SROC/AUC) or conclusions reported in the manuscript.

Rationale for removing AUC (per study) rather than “Method”: The per-study AUC is often not directly comparable across studies because many papers report single-threshold accuracy (one sensitivity/specificity pair) rather than a full ROC, and AUCs may be estimated under differing assumptions. Since our quantitative synthesis relies on the bivariate model of sensitivity and specificity and presents the pooled SROC/AUC, listing per-study AUCs in the main table is redundant and can be misleading with heterogeneous thresholds. By contrast, “Method” (e.g., RT-qPCR, microarrays, NGS) describes the index test characteristics, which are essential for assessing applicability and sources of heterogeneity per PRISMA-DTA/Cochrane DTA guidance.

Methodological differences in the index test were pre-specified and used in our subgroup analyses and meta-regressions; removing this column would hinder interpretability and reproducibility for readers.

For these reasons, we believe adding Total (N) and keeping Method in the main table—while relocating per-study AUCs to the Supplement—offers the clearest and most standards-aligned presentation.

6. Results section, “Subgroup Analysis and Meta-Regression in CTL vs DR, T2DM vs DR, and NPDR vs PDR Comparisons” subsection (pdf file, revised manuscript, page 66, lines 308-309): The authors assert that “studies conducted in China demonstrated higher diagnostic performance across all three comparison groups, as reflected by AUC.” I do not know if this is the case for CTL vs DR and T2DM vs DR. In the comparison between CTL and DR (Table 2), only one study was not performed in China, which hinders its interpretation. Regarding the comparison between T2DM and DR (Table 3), the difference in the AUC value was 0.01.

We thank the reviewer for this careful observation. In the Results section our intention was descriptive. The subgroup AUCs are:

• CTL vs DR: China 0.86 vs non-China 0.56

• T2DM vs DR: China 0.87 vs non-China 0.86 (ΔAUC = 0.01)

• NPDR vs PDR: China 0.90 vs non-China 0.87

We agree that interpretation is limited by the geographical imbalance, particularly in CTL vs DR, where only one study was conducted outside China, and that for T2DM vs DR the AUCs are essentially comparable.

This limitation is also explicitly acknowledged in the Discussion, where we note that the predominance of Chinese cohorts may introduce demographic bias and limit generalizability to other populations, underscoring the need for broader geographic representation in future studies:

“A predominant number of the included studies were conducted in Chinese populations, which may introduce demographic bias and limit the generalizability of the findings to other ethnic groups, particularly Western cohorts. While this does not compromise internal validity, it underscores the need for broader geographic representation in future studies.”

7. Results section (pdf file, revised manuscript, page 70, lines 371-376): The text refers to the paper by Shaker et al. (reference #26) as if there were more than one study: “Cook’s distance analysis pinpointed 371 Shaker et al. (miR-20b and miR-17-3p)(26) as influential studies (Figure 7G). Yet, removing these studies caused only marginal changes: pooled sensitivity declined from 81.0% to 78.9%, and specificity increased from 80.0% to 81.1%. Interestingly, heterogeneity (I²) rose notably from 74.12% to 97.71%, suggesting that these studies contributed to controlled variance across studies. Outlier assessment confirmed no excessive deviations (Figure 7H).” Is it correct (“two studies”)?

We appreciate this observation and agree that our wording could be misread. Shaker et al. is a single article, but it reports two distinct index tests (miR-20b and miR-17-3p), each with its own diagnostic accuracy metrics. In diagnostic test accuracy meta-analyses, when a paper evaluates multiple biomarkers with separate accuracy data, each biomarker is extracted as an independent DTA entry to avoid aggregation bias. This is reflected in Table 1, where certain authors appear more than once because different miRNAs were assessed, each contributing a separate 2×2 dataset.

To eliminate ambiguity, we have revised the Results wording as follows: “Cook’s distance analysis identified two influential miRNA-specific entries from Shaker et al. (miR-20b and miR-17-3p) (26) (Fig. 7G). Removing these two miRNAs led to only marginal changes: pooled sensitivity declined from 81.0% to 78.9% and specificity increased from 80.0% to 81.1%; I² rose from 74.12% to 97.71%, indicating these entries contributed to controlled between-study variance. Outlier assessment confirmed no excessive deviations (Fig. 7H).”

8. Regarding I² (heterogeneity estimates), is it really necessary to use two decimal places to express this metric (Results section)?

This point is well taken. To ensure consistent, reader-friendly formatting, we now report I² as an integer percentage (no decimals) throughout the manuscript. For the same reason, summary sensitivity and specificity are presented as whole percentages, with the accompanying 95% CIs providing the formal measure of precision. Readers who need finer granularity can consult the forest plots/SROC panels and the supplementary dataset, where the underlying exact values are available.

In addition, to avoid redundancy and reduce visual clutter in the subgroup Results text, we have removed inline 95% confidence intervals for subgroup metrics (sensitivity, specificity, DOR, etc.). These CIs are reported consistently and in full in the corresponding tables (Tables 2–4) and the Supplementary dataset, which serve as the canonical source for detailed numerical results. This change streamlines the narrative while preserving all necessary precision for interested readers.

Also, We have added a statement to the Statistical Analysis section clarifying that percentages are rounded for clarity and consistency. The text reads:

“Percentages (e.g., sensitivity, specificity, and I²) are reported as whole numbers rounded to the nearest integer; exact estimates and 95% confidence intervals are provided in the corresponding tables and figures.”

9. Limitations section (pdf file, revised manuscript, page 75, lines 481-483):“Fourth, pre-analytical and analytical heterogeneity was substantial across studies. Variations in the biological sample type, normalization strategies, and miRNA isolation and detection platforms contributed significantly to inconsistency in results.” It is known that these factors affect the quantification of miRNA expression. But are the authors referring to the results of their analyses? Does “detection platforms” mean the method of quantifying microRNAs? I did not find any mention to the miRNA isolation method used in the previous studies in the manuscript. Moreover, as per Table 1 and S3 Table, all but one study measured the miRNA levels by RT-qPCR. In sum, did the authors perform the meta-analysis considering these two factors (miRNA isolation and detection platforms)?

Thank you for raising this point. We confirm that the limitation refers to our own findings and to the variables we analyzed/explored for heterogeneity.

• What “detection platforms” means: it denotes the miRNA quantification method (e.g., RT-qPCR vs RNA-seq). In our dataset, 15/16 studies used RT-qPCR and 1 used RNA-seq (Table 1). Because of this near-absence of contrast, platform could not be modeled as a moderator without producing unstable estimates; hence it is discussed as a potential source of heterogeneity and listed as a limitation.

• What we did analyze: we pre-specified and modeled specimen type and normalization strategy, and they indeed showed effects in subgroup analyses and/or meta-regressions (Tables 2–4; Fig. 6). For example, normalization strategy was significantly associated with sensitivity in all three comparisons in the meta-regression, and sp

---

## [Decision Letter · Decision Letter 2]

24 Sep 2025

Dear Dr. Oltra,

Thank you for submitting your manuscript to PLOS ONE. After careful consideration, we feel that it has merit but does not fully meet PLOS ONE’s publication criteria as it currently stands. Therefore, we invite you to submit a revised version of the manuscript that addresses the points raised during the review process.

We look forward to receiving your revised manuscript.

Kind regards,

Yalong Dang

Academic Editor

PLOS ONE

Journal Requirements:

Reviewers' comments:

Reviewer's Responses to Questions

**Comments to the Author**

Reviewer #5: (No Response)

2. Is the manuscript technically sound, and do the data support the conclusions?

Reviewer #5: Yes

3. Has the statistical analysis been performed appropriately and rigorously?

Reviewer #5: I Don't Know

4. Have the authors made all data underlying the findings in their manuscript fully available?

Reviewer #5: Yes

5. Is the manuscript presented in an intelligible fashion and written in standard English?

Reviewer #5: Yes

Reviewer #5: The manuscript was revised to address the comments raised in the previous submission (R1). However, the submitted manuscript (R2) does not present some of the modifications stated in the rebuttal letter and some errors (that did not exist) were introduced into the new version of the manuscript (PONE-D-25-27197_R2.pdf file). After I started writing my comments, I found out that the “clean manuscript” (pages 15 to 39) is not identical to the “manuscript with track changes” (pages 48 to 73), which makes it difficult to follow what has been modified (or not) in the revised manuscript. The comments I provide below are based on the manuscript included on pages 48 to 78. Finally, I have to say that I did not check the figures and the supplementary tables in the revised manuscript.

1) Abstract section (revised manuscript with track changes, page 48):

In the response to the comment #1a, the authors state in the rebuttal letter that they “have revised the first sentence of the Abstract accordingly to improve clarity and avoid redundancy. The updated sentence now reads: “Purpose: To evaluate the diagnostic accuracy of circulating miRNAs in distinguishing between different diabetic retinopathy stages in type 2 diabetes mellitus.”

Although this detail does not compromise the manuscript, the purpose remains the same. Instead, the second sentence (first one of the Methods) was modified (revised manuscript with track changes, page 48, lines 18-20).

2) Abstract section (revised manuscript with track changes, page 48):

An error was introduced in the sentence “We searched PubMed, CENTRAL, Scopus, Web of Science, ScienceDirect and ClinicalTrials (up to 20 January 2025) for diagnostic test accuracy studies […]” (lines 21-27) by deleting the following terms: “type 2 diabetes mellitus without retinopathy versus diabetic retinopathy”. Now, the sentence is incomplete.

3) Introduction section (revised manuscript with track changes, page 49):

In response to the comment #2, the reference by Lundeen et al. was replaced by the paper by Thomas et al. However, the study by Lundeen et al. remains in the Bibliography (page 70) as the reference #2. Therefore, all references from Thomas et al. onward should be renumbered.

4) Results section (revised manuscript with track changes, page 56):

In response to the comment #5, the authors added a column with the total N in Table 1 described on page 22, but not on that presented on page 56. I think this is not a problem, as the Journal uses the clean version of the manuscript to typeset the proof.

5) Results section (revised manuscript with track changes, page 57):

In response to the comment #8, the authors decided to report I² as an integer percentage (throughout the manuscript). However, the following sentence still has the I² estimates with two decimals between the parentheses and needs to be corrected (lines 203-206): “The pooled estimates from the random‐effects model showed a summary sensitivity of 77% (70–82), with an I² of 47% (1.39–92.75), and a summary specificity of 84% (77–89), with an I² of 62% (31.26–93.30), indicating moderate heterogeneity (Fig 3A-B).”

6) Editing and typo errors:

In response to the comment #10m, the authors added the following sentences to the Methods section: “The complete extraction dataset is provided in S3 Table.” and “Additional details are provided in S4, S5, and S6 tables.” These additions were made to the clean manuscript (pages 19 and 21), but not to the manuscript with track changes.

7) Funding section:

In the response to the comment #1b, the authors state in the rebuttal letter that “no funds from these grants were specifically allocated or used for the conduct of this systematic review and meta-analysis. To avoid any possible misunderstanding, we have removed the Funding section from the manuscript and clarified that this work received no dedicated external financial support.”

However, the Funding section was neither removed from the manuscript nor corrected according to this statement (revised manuscript with track changes, page 69).

**Do you want your identity to be public for this peer review?** For information about this choice, including consent withdrawal, please see our Privacy Policy

Reviewer #5: **Yes: ** Kátia Gonçalves dos Santos

---

## [Author Response · Author response to Decision Letter 3]

29 Sep 2025

Reviewer #5: The manuscript was revised to address the comments raised in the previous submission (R1). However, the submitted manuscript (R2) does not present some of the modifications stated in the rebuttal letter and some errors (that did not exist) were introduced into the new version of the manuscript (PONE-D-25-27197_R2.pdf file). After I started writing my comments, I found out that the “clean manuscript” (pages 15 to 39) is not identical to the “manuscript with track changes” (pages 48 to 73), which makes it difficult to follow what has been modified (or not) in the revised manuscript. The comments I provide below are based on the manuscript included on pages 48 to 78. Finally, I have to say that I did not check the figures and the supplementary tables in the revised manuscript.

1) Abstract section (revised manuscript with track changes, page 48): In the response to the comment #1a, the authors state in the rebuttal letter that they “have revised the first sentence of the Abstract accordingly to improve clarity and avoid redundancy. The updated sentence now reads: “Purpose: To evaluate the diagnostic accuracy of circulating miRNAs in distinguishing between different diabetic retinopathy stages in type 2 diabetes mellitus.” Although this detail does not compromise the manuscript, the purpose remains the same. Instead, the second sentence (first one of the Methods) was modified (revised manuscript with track changes, page 48, lines 18-20).

We appreciate your observation. Upon re-evaluating the Purpose and Methods sections, we noticed that they could lead to ambiguity due to a repetition of the study objective. Accordingly, we have reformulated both sections to avoid redundancy and improve clarity in abstract section.

2) Abstract section (revised manuscript with track changes, page 48): An error was introduced in the sentence “We searched PubMed, CENTRAL, Scopus, Web of Science, ScienceDirect and ClinicalTrials (up to 20 January 2025) for diagnostic test accuracy studies […]” (lines 21-27) by deleting the following terms: “type 2 diabetes mellitus without retinopathy versus diabetic retinopathy”. Now, the sentence is incomplete.

We understand the reviewer’s concern. However, we would like to clarify that this was an intentional choice to avoid redundancy. In the Abstract (lines 17–18), we had already introduced the full terms and abbreviations for diabetic retinopathy (DR) and type 2 diabetes mellitus (T2DM). Repeating these abbreviations immediately in the following sentence would unnecessarily lengthen the Abstract without adding clarity. By maintaining the current formulation, the meaning remains fully understandable while ensuring conciseness, which is essential for the Abstract.

3) Introduction section (revised manuscript with track changes, page 49):

In response to the comment #2, the reference by Lundeen et al. was replaced by the paper by Thomas et al. However, the study by Lundeen et al. remains in the Bibliography (page 70) as the reference #2. Therefore, all references from Th

We appreciate the reviewer’s observation. The discrepancy arose because the EndNote bibliography was not fully updated in the track-changes version of the manuscript. However, the correction was made in the clean version, where the reference list has been properly updated and renumbered. We regret the oversight and confirm that the final revised manuscript reflects the correct reference Thomas et al. onward should be renumbered.

4) Results section (revised manuscript with track changes, page 56): In response to the comment #5, the authors added a column with the total N in Table 1 described on page 22, but not on that presented on page 56. I think this is not a problem, as the Journal uses the clean version of the manuscript to typeset the proof.

Thank you for this observation. We had not realized that while the updated version of Table 1 (including the total N column) was uploaded as an independent file, the same correction was not applied in the manuscript version. We have now corrected this inconsistency and ensured that the revised manuscript includes the updated Table 1

5) Results section (revised manuscript with track changes, page 57): In response to the comment #8, the authors decided to report I² as an integer percentage (throughout the manuscript). However, the following sentence still has the I² estimates with two decimals between the parentheses and needs to be corrected (lines 203-206): “The pooled estimates from the random‐effects model showed a summary sensitivity of 77% (70–82), with an I² of 47% (1.39–92.75), and a summary specificity of 84% (77–89), with an I² of 62% (31.26–93.30), indicating moderate heterogeneity (Fig 3A-B).”

We agree with the reviewer’s observation and realized that in some parts of the manuscript we reported the confidence intervals for heterogeneity (I²), while in others we did not. To avoid confusion, we have removed the confidence intervals for I², as the most relevant information lies in reporting the confidence intervals for sensitivity and specificity. Moreover, the forest plot figures already provide the confidence intervals for I², should readers wish to consult them. We have applied the same approach to the reporting of confidence intervals for sensitivity, specificity, and DOR in the section Subgroup Analysis and Meta-Regression in CTL vs DR, T2DM vs DR, and NPDR vs PDR Comparisons. There are three tables that clearly present all confidence intervals in detail. Including excessive numerical information in the text may hinder the readability of the results. Furthermore, we have observed that other scientific articles follow a similar approach in order to highlight their key findings in the subgroup analyses.

6) Editing and typo errors: In response to the comment #10m, the authors added the following sentences to the Methods section: “The complete extraction dataset is provided in S3 Table.” and “Additional details are provided in S4, S5, and S6 tables.” These additions were made to the clean manuscript (pages 19 and 21), but not to the manuscript with track changes.

We regret the confusion. These sentences have now been properly added in the revised manuscript with track changes, so that both versions are consistent.

7) Funding section: In the response to the comment #1b, the authors state in the rebuttal letter that “no funds from these grants were specifically allocated or used for the conduct of this systematic review and meta-analysis. To avoid any possible misunderstanding, we have removed the Funding section from the manuscript and clarified that this work received no dedicated external financial support.” However, the Funding section was neither removed from the manuscript nor corrected according to this statement (revised manuscript with track changes, page 69).

The reviewer is correct, and we apologize for this misunderstanding. Since several authors were revising the manuscript simultaneously, an earlier version was mistakenly used as the basis for further edits, which led to the Funding section not being updated as intended. This error has now been corrected, and the revised manuscript accurately reflects the statement provided in our rebuttal.

---

## [Decision Letter · Decision Letter 3]

12 Oct 2025

Circulating microRNAs as biomarkers for diabetic retinopathy stage identification: a DTA systematic review and meta-analysi s

PONE-D-25-27197R3

Dear Dr. Oltra,

We’re pleased to inform you that your manuscript has been judged scientifically suitable for publication and will be formally accepted for publication once it meets all outstanding technical requirements.

Kind regards,

Yalong Dang

Academic Editor

PLOS ONE

Additional Editor Comments (optional):

Reviewers' comments:

Reviewer's Responses to Questions

**Comments to the Author**

Reviewer #5: All comments have been addressed

2. Is the manuscript technically sound, and do the data support the conclusions?

Reviewer #5: Yes

3. Has the statistical analysis been performed appropriately and rigorously?

Reviewer #5: I Don't Know

4. Have the authors made all data underlying the findings in their manuscript fully available?

Reviewer #5: Yes

5. Is the manuscript presented in an intelligible fashion and written in standard English?

Reviewer #5: Yes

Reviewer #5: All the remaining inconsistencies in the manuscript were solved. The only thing I noticed on a quick revision is that the following sentence is now duplicated in the Abstract section: “Data were synthesized via a bivariate random-effects meta-analysis, with subgroup analyses, meta-regression, and sensitivity tests to explore heterogeneity”.

**Do you want your identity to be public for this peer review?** For information about this choice, including consent withdrawal, please see our Privacy Policy

Reviewer #5: **Yes: ** Kátia Gonçalves dos Santos

---

## [Editor Report · Acceptance letter]

PONE-D-25-27197R3

PLOS ONE

Dear Dr. Oltra,

I'm pleased to inform you that your manuscript has been deemed suitable for publication in PLOS ONE. Congratulations! Your manuscript is now being handed over to our production team.

Kind regards,

on behalf of

Dr Yalong Dang

Academic Editor

PLOS ONE